# Engineering and application of a biosensor with focused ligand specificity

Dennis Della Corte[1,2], Hugo L. van Beek[3], Falk Syberg[4], Marcus Schallmey [3], Felix Tobola [3], Kai U. Cormann[3], Christine Schlicker[4], Philipp T. Baumann[3], Karin Krumbach[3], Sascha Sokolowsky[3], Connor J. Morris [2], Alexander Grünberger [3,5], Eckhard Hofmann [4], Gunnar F. Schröder [1,6,7] & Jan Marienhagen [3,8✉]

Cell factories converting bio-based precursors to chemicals present an attractive avenue to a sustainable economy, yet screening of genetically diverse strain libraries to identify the best-performing whole-cell biocatalysts is a low-throughput endeavor. For this reason, transcriptional biosensors attract attention as they allow the screening of vast libraries when used in combination with fluorescence-activated cell sorting (FACS). However, broad ligand specificity of transcriptional regulators (TRs) often prohibits the development of such ultra-high-throughput screens. Here, we solve the structure of the TR LysG of *Corynebacterium glutamicum*, which detects all three basic amino acids. Based on this information, we follow a semi-rational engineering approach using a FACS-based screening/counterscreening strategy to generate an L-lysine insensitive LysG-based biosensor. This biosensor can be used to isolate L-histidine-producing strains by FACS, showing that TR engineering towards a more focused ligand spectrum can expand the scope of application of such metabolite sensors.

[1] Institute of Biological Information Processing, IBI-7: Structural Biochemistry, Forschungszentrum Jülich, D-52425 Jülich, Germany. [2] Department of Physics & Astronomy, Brigham Young University, Provo, UT, USA. [3] Institute of Bio- and Geosciences, IBG-1: Biotechnology, Forschungszentrum Jülich, D-52425 Jülich, Germany. [4] Protein Crystallography, Biophysics, Ruhr University Bochum, D-44780 Bochum, Germany. [5] Multiscale Bioengineering, Bielefeld University, D-33615 Bielefeld, Germany. [6] Department of Physics, Heinrich-Heine University, D-40225 Düsseldorf, Germany. [7] Jülich Centre for Structural Biology (JuStruct), Forschungszentrum Jülich, D-52425 Jülich, Germany. [8] Institute of Biotechnology, RWTH Aachen University, Worringer Weg 3, D-52074 Aachen, Germany. ✉email: j.marienhagen@fz-juelich.de

For decades, microorganisms have been successfully transformed into cell factories for sustainable synthesis of industrially useful products in sectors, such as pharmaceuticals, food, feed, chemicals, detergents, and biofuels[1]. With a growing world population and climate change creating a scarcity of water, land, and other vital resources, industrial biotechnology can make a decisive contribution toward meeting these global challenges. However, an envisioned bio-based future and sustainable economy requires development of efficient microbial production strains for a multitude of small molecules and applications. Whereas generation of genetic diversity (random or rational), as well as high-throughput genetic engineering of microorganisms does not pose a problem in this context, the rapid evaluation of a large number of clones is still challenging[2]. In principle, each genetic variant must be cultivated and evaluated for its productivity individually, requiring costly and low-throughput methods, such as chromatography or mass spectrometry.

Biosensors have emerged as valuable tools for strain engineering and have changed the manner and speed, in which production strains are developed[3–5]. In general, biosensors detect changes in (intracellular) concentrations of small compounds and translate this input into a genetic output. Thus, biosensors can be used as molecular switches for rerouting microbial metabolism in response to the presence of a certain molecule. Alternatively, intracellular accumulation of a small molecule of interest can be converted into a machine-readable output, such as fluorescence. This allows for the rapid ultra-high-throughput screening of vast libraries of genetically diverse microorganisms at the single-cell level when combined with fluorescence-activated cell sorting (FACS), enabling the analysis of >$10^4$ variants per second[6]. The powerful FACS approach renders costly individual cultivation and evaluation of all clones in a given library unnecessary, and thus significantly speeds up design–build–test cycles. For such applications, RNA-based and transcription factor-based biosensors have been described in literature[7]. In particular, transcription factor-based biosensors, comprised of a fluorescent protein-encoding reporter gene whose expression is controlled by a ligand-inducible transcriptional regulator (TR), have been successfully used for strain engineering in both, academic and industrial settings[8–10].

Furthermore, ligand-binding properties of several TRs can be modified by protein engineering, yielding custom-made biosensors for detecting different compounds of interest[11]. A well-known example is the transcriptional activator AraC, detecting L-arabinose, which was evolved toward novel ligand specificities for D-arabinose[12], mevalonate[13], and recently triacetic acid lactone[14]. More recently, the repressor of the lac operon, LacI, was engineered to also accept D-fucose, lactitol, sucralose, or gentiobiose[15]. CatM of *Acinetobacter baylyi* ADP1, involved in the degradation of aromatic compounds, was also altered to accept benzoate[16]. However, only a few of these adapted biosensors were used for larger-scale screening campaigns, probably because biosensor properties, such as weak ligand binding or low reporter gene expression prohibited implementation of FACS-based screenings[9].

A major challenge that has not been tackled yet is the relaxed ligand specificity of some TRs, limiting their efficient utilization in a biosensor system. A well-known example is the TR LysG of the amino acid producer *Corynebacterium glutamicum* ATCC 13032. LysG detects the basic amino acids L-lysine, L-histidine, and L-arginine and activates gene expression of the amino acid transporter encoding gene *lysE* in the presence of elevated intracellular amino acid concentrations[17]. The biosensor pSenLys, comprised of LysG and its target promoter controlling *lysE* expression, proved to be a valuable tool for identifying mutagenic

hot spots contributing to L-lysine synthesis in the genome of *C. glutamicum* (Fig. 1a)[6]. However, all attempts to use this biosensor for engineering *C. glutamicum* toward overproducing the biotechnologically interesting amino acids L-histidine or L-arginine failed, and only L-lysine-accumulating variants could be identified in randomly mutagenized libraries. Only when key genes of L-arginine- or L-histidine biosynthesis were subjected to random mutagenesis prior to FACS screening using pSenLys could *C. glutamicum* variants accumulating these two basic amino acids be isolated[18]. The reason for this observation is still unclear, possibly the dissociation constant ($K_D$) for L-histidine and L-arginine is too high compared to the intracellular abundance of these amino acids, making changes in intracellular concentrations of these amino acids difficult to detect. Furthermore, as the network of biosynthetic pathways contributing to L-histidine and L-arginine synthesis are more complex and subject to a more stringent metabolic control compared to L-lysine[19], beneficial mutations leading to L-histidine or L-arginine accumulation are less likely to occur and therefore more difficult to identify. Either way, reduced affinity of LysG to L-lysine would allow for identifying L-histidine- and L-arginine- producing strains using a biosensor-based FACS-screening strategy.

In this study, we present a detailed structural and biochemical characterization of the TR LysG with regard to its binding properties. Based on this information, we engineer LysG semi-rationally toward no binding of L-lysine, while maintaining its L-histidine- and L-arginine-binding capabilities, and employ molecular dynamics (MD) simulations to understand the underlying structure–function relationship of a crucial single amino acid substitution. The engineered LysG variant with the narrowed ligand spectrum is subsequently used to construct the biosensor pSenHis. As part of this sensor, the TR stimulates activity of a promoter that drives the synthesis of the fluorescent protein EYFP (enhanced yellow fluorescent protein), which in turn allows FACS of individual cells. This biosensor is studied on the single-cell level using microfluidics and successfully applied in a FACS-based screening of $10^7$ chemically mutagenized *C. glutamicum* wild-type cells for identifying L-histidine-producing *C. glutamicum* variants. We believe that this approach holds the promise to serve as guideline for biosensor reengineering toward tailor-made properties for a multitude of possible applications.

## Results

### A crystallographic structure model of LysG of *C. glutamicum*.
Engineering of ligand-binding properties of a TR requires detailed knowledge of its structure, in particular its ligand-binding site, in order to identify important amino acid residues involved in effector binding. LysG is a member of the LysR-type transcriptional regulator (LTTR) family, a large and highly conserved group of TRs ubiquitous in bacteria[20]. As no structure of LysG of *C. glutamicum* was available, we set out to solve the structure of full-length LysG with and without basic amino acids as ligands by single-crystal X-ray crystallography to obtain a sound experimental basis for analysis of effector binding. Interestingly, all attempts to crystallize LysG in the presence of L-lysine and L-histidine failed, but diffraction datasets could be collected to solve the structure of LysG in the free form [LysG], and together with bound ligand L-arginine [LysG + Arg] at 2.52 Å and 3.00 Å, respectively (Supplementary Table 1).

LysG shows the typical features of LTTRs, a two-lobed regulatory domain linked to the DNA-binding domain by a flexible loop (Fig. 1b and Supplementary Fig. 1). Two LysG monomers are coupled by dimerization of their DNA-binding domains, forming a winged helix-turn-helix motif responsible for DNA binding. As is the case for all known LTTR structures, one

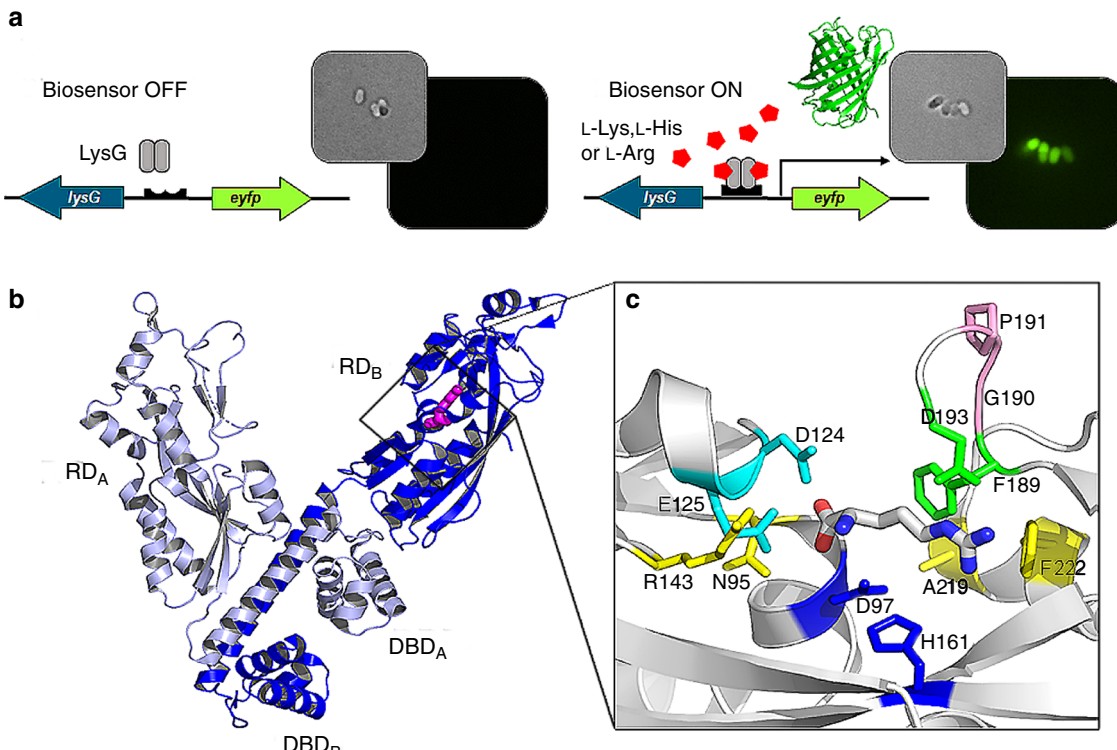

**Fig. 1 Functional principle of the biosensor pSenLys and structure of its TR-component LysG. a** Schematic representation of the pSenLys biosensor for the intracellular detection of basic amino acids in *C. glutamicum* and phase-contrast/fluorescence microscopy images of *C. glutamicum* cells carrying pSenLys. In the presence of elevated intracellular concentrations of any of the three basic amino acids L-lysine, L-histidine, or L-arginine, the transcriptional activator LysG binds the respective inducer amino acid and activates expression of the reporter gene *eyfp*. As a result, cells of *C. glutamicum* show fluorescence. **b** Cartoon representation of the LysG homodimer in complex with the effector L-arginine. The DNA-binding domains (DBD) of the compact (light blue) and extended (blue) protomers dimerize to form a winged helix-turn-helix motif responsible for DNA binding. L-arginine bound in the regulatory domain (RD) of the extended protomer is shown as pink stick model. **c** Coordination of L-arginine in the ligand-binding site of the RD. The RD is shown in cartoon representation; bound L-arginine and selected residues of the binding pocket later targeted for pairwise mutation are shown in stick representation.

monomer adopts an extended conformation, whereas the second monomer arranges in a more compact conformation. During size-exclusion chromatography, LysG eluted as a peak representing a molecular mass of 158 kDa. Based on the mass of 34 kDa deduced from the protein sequence, this indicates a tetrameric assembly of LysG in solution. The ligand-binding pocket is formed by the interface of the two lobes of the regulatory domain. Binding of L-arginine in the extended protomer of [LysG + Arg] leads to a small but substantial tilt of helix 6 by 8° compared to the conformation in the ligand-free protomer [LysG], which enables coordination of the ligand L-arginine by amino acid residues D124 and E125 in the [LysG + Arg] structure (Supplementary Fig. 2). Within the ligand-binding cavity of the [LysG + Arg] structure, sidechains of eight residues (N95, D97, D124, E125, H161, F189, D193, and F222) are located within a distance of <3.6 Å to the bound L-arginine molecule, potentially interacting with this effector molecule and the other two basic amino acids (Fig. 1c). However, since no diffracting LysG crystals in the presence of L-lysine or L-histidine could be obtained, blind docking calculations using the [LysG + Arg] structure and AutoDock Vina[21] were performed, which predicted the positions of L-arginine, L-lysine, and L-histidine to be in the same binding pocket, with a maximum center of mass deviation of the top ten docking poses from the crystal L-arginine of 1.2 Å, 1.2 Å, and 1.3 Å, respectively.

**Structure-guided engineering of LysG.** Based on this structural data, 12 amino acid positions in LysG were targeted for site-

saturation mutagenesis with the aim to screen the resulting libraries for LysG variants with reduced L-lysine-binding capabilities. In addition to eight residues (N95, D97, D124, E125, H161, F189, D193, and F222) with direct contact to the amino acid ligands, four second shell residues were included, which are not strictly conserved in LTTR-type TRs (R143, A219, G190, and P191). R143 is located in the regulatory domain, A219 is positioned at the bottom of the ligand-binding pocket, and G190 and P191 in the loop adjacent to F189 (Fig. 1c). These 12 residues were mutated pairwise (N95 + R143; D97 + H161; D124 + E125; F189 + D193; G190 + P191; and A219 + F222) by multi-site-directed saturation mutagenesis to increase the protein sequence space and to directly take advantage of potentially synergistic effects of neighboring amino acid substitutions. The resulting *lysG* variants were subcloned into the biosensor plasmid pSenLys, thereby replacing the wild-type *lysG* gene. The six plasmid libraries containing 6,000–12,000 variants each were introduced into *C. glutamicum* Δ*lysEG*, devoid of the genome-encoded *lysG* and *lysE* genes. These two genes were deleted to prevent biosensor activation by wild-type LysG activity and to ensure that only the plasmid-based *lysE* promoter controlling *eyfp* expression is targeted. Subsequently, a FACS-based screen for identifying L-histidine-specific biosensor variants was conducted. For this purpose, all six biosensor libraries were combined (>52,000 variants) and subjected to a FACS-based five-step screening/counterscreening strategy for identifying L-lysine-insensitive biosensor variants, still capable of *eyfp* reporter gene expression upon recognition of L-histidine (Fig. 2a). During this process, positive screening steps in the presence of 3 mM L-His-L-Aladipeptides for

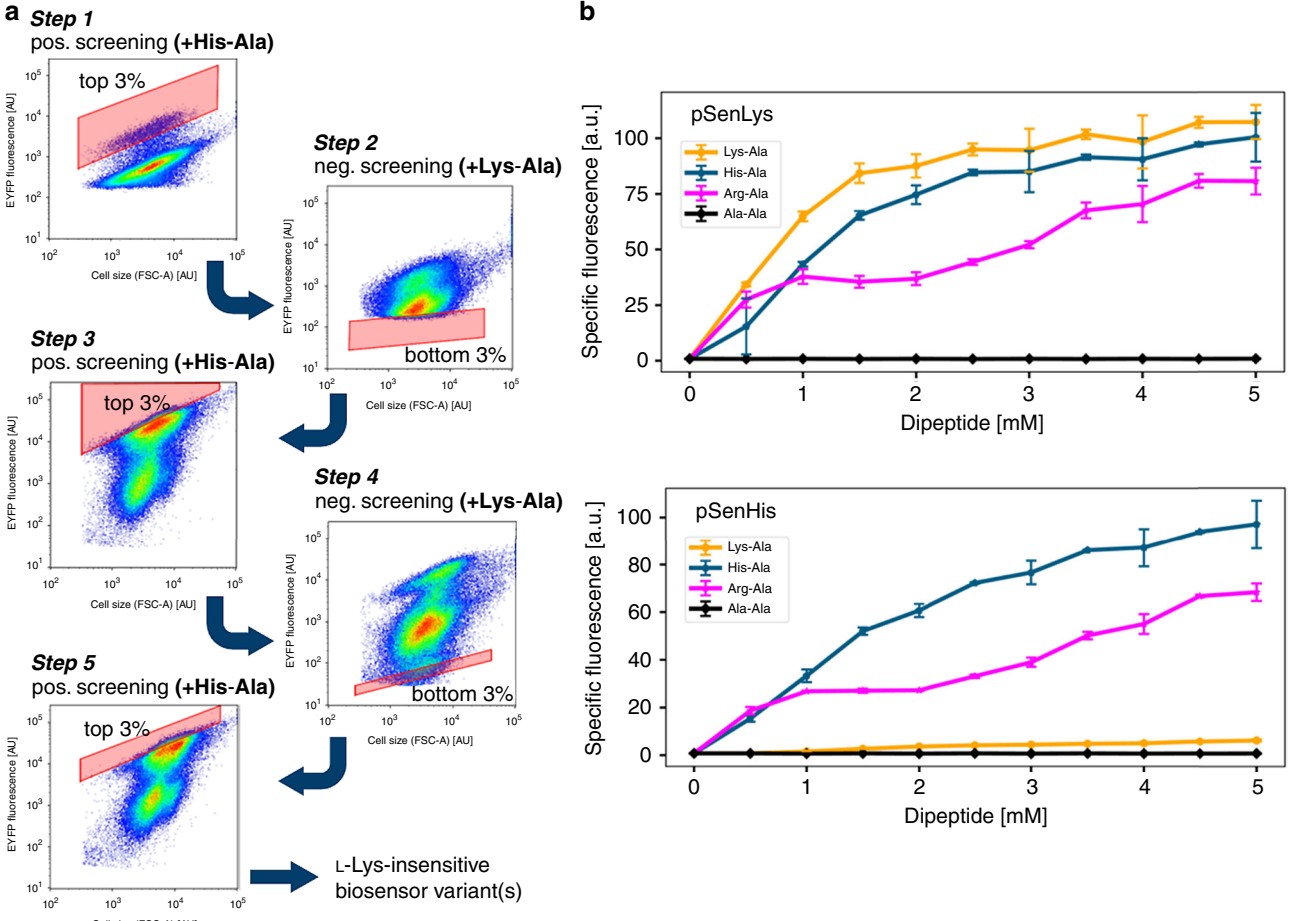

**Fig. 2 FACS-based positive/negative screening to identify an L-lysine-insensitive biosensor. a** FACS plots obtained during the five-step screening/counterscreening campaign for identifying L-lysine-insensitive L-histidine biosensor variants. During this procedure, fluorescent cells responding to L-His-L-Ala dipeptides were collected during positive screening, alternating with a collection of nonfluorescent cells during negative screening in the presence of L-Lys-L-Ala dipeptides. During the last positive screening, 96 individual clones were sorted for further characterization. **b** Initial characterization of the identified L-lysine-insensitive biosensor variant. Fluorescence response of the biosensors pSenLys (top) and pSenHis (bottom) to the presence of various dipeptides at different concentrations during microtiter plate cultivations. In each case, the specific fluorescence as ratio of the fluorescence determined after 22 h over the culture backscatter (as measure for cell density) is shown. All data represent mean values from three biological replicates including standard deviations.

identifying L-histidine-responsive biosensor variants (isolation of top 3% fluorescing cells) alternated with negative screening steps in the presence of 3 mM L-Lys-L-Ala dipeptides, in which still-L-lysine-responsive biosensors were discarded (isolation of bottom 3% non-fluorescent cells). Notably, supplementation of dipeptides was preferred over addition of single amino acids in these experiments as dipeptides are more readily taken up (and hydrolyzed) by *C. glutamicum*[6].

During this screening process, L-histidine-responsive but L-lysine-insensitive biosensor variants could be successfully enriched, and 96 clones showing high fluorescence in the presence of L-His-L-Ala dipeptides were individually collected in a 96-well plate. During microtiter plate-based rescreening, 71 clones turned out to be viable, with 14 clones showing a high specific L-histidine/L-lysine fluorescence ratio compared to the parent pSenLys biosensor with the wild-type LysG variant. DNA sequencing of the 14 biosensor variants revealed that all *lysG* variants carried the same two single point mutations in the open reading frame: C656T and T666C. Whereas the T666C transition was silent (F222F), the C656T transition did cause an A→L amino acid substitution at position 219. A control experiment, in which the C656T transition was introduced into the parent plasmid carrying the wild-type *lysG* gene by site-directed

mutagenesis, confirmed the results obtained and showed that this single amino acid substitution is indeed responsible for the observed biosensor phenotype.

**The pSenHis biosensor is L-lysine-insensitive**. The specific fluorescence of the engineered pSenLys-A219L sensor variant (hereafter named pSenHis) and the parent sensor pSenLys in response to varying L-His-L-Ala, L-Lys-L-Ala and L-Arg-L-Ala dipeptide concentrations was recorded during biolector cultivations for a more detailed characterization (Fig. 2b). As control experiments to reveal potential background fluorescence in the OFF-state of the biosensor, cultivations in the presence of different L-Ala-L-Ala dipeptide concentrations and in the absence of all dipeptides were also performed. Here, pSenHis showed a drastically reduced response to L-lysine in comparison to pSenLys, whereas the response to L-histidine resembles that of the parent biosensor. Interestingly, background fluorescence in the presence of L-alanine and in the absence of any amino acid at all was also reduced by >15% when *C. glutamicum* Δ*lysEG* carried pSenHis instead of pSenLys. This difference leads to a higher fold response in fluorescence between ON- and OFF-state, and hence might allow for a better separation of positive and negative clones

during FACS-based screenings employing this biosensor. In comparison to pSenLys, the fluorescence response of pSenHis to L-arginine was slightly reduced, although the parameter L-arginine specificity was not included in the screening/counter-screening procedure leading to the isolation of pSenHis.

In addition to biolector cultivations only allowing for recording of an average fluorescence response of whole populations, spatiotemporal microfluidic single-cell analyses using microfabricated organosilicon chips was performed. These experiments enabled cultivation of *C. glutamicum ΔlysEG* cells carrying pSenLys or pSenHis, providing quantitative data on fluorescence of individual cells in the microcolony in response to supplemented dipeptides, using automated time-lapse microscopy under well-defined environmental conditions (Fig. 3, Supplementary Fig. 3 and Supplementary Movies 1–8). Without supplementation of any dipeptides, growth of both strains was uniform, and no fluorescence could be detected (Fig. 3 and Supplementary Fig. 3). In the presence of L-His-L-Ala or L-Lys-L-Aladipeptides, *C. glutamicum ΔlysEG* pSenLys cells showed a homogenous fluorescence response to supplementation of both dipeptides, with a broader distribution of fluorescence in the case of L-lysine. During microfluidic cultivations with *C. glutamicum ΔlysEG* cells carrying the engineered pSenHis biosensor bearing the A219L amino acid substitution, a different response could be detected.

Whereas the pSenHis biosensor response to L-histidine resembled that of pSenLys, no cell showed any response to L-lysine (Fig. 3). Only very low background fluorescence, negligibly higher compared to the negative control without any dipeptide supplementation, could be detected. Interestingly, supplementation of L-Arg-L-Ala dipeptides yielded very heterogeneous microcolonies with regard to fluorescence for both biosensors (Supplementary Fig. 3). This effect could be connected to the biphasic biosensor response to L-Arg-L-Ala dipeptides observed in biolector cultivations (Fig. 2b), and could hint at elevated intracellular L-arginine concentrations affecting cell growth. Addition of L-Ala-L-Ala dipeptides as second negative control matched the results obtained during biolector cultivations, since only a very low fluorescence could be detected for cells carrying pSenLys, but no fluorescence at all for pSenHis (Supplementary Fig. 3).

**Substitution A219L affects ligand interaction and dynamics.** Interaction of basic amino acids with LysG was analyzed with isothermal titration calorimetry (ITC) and MD simulations. ITC with purified TR proteins was performed to quantify and compare ligand-binding properties of LysG and the engineered LysG-A219L variant. The highest affinity of the parent TR LysG could be determined for L-histidine ($K_D = 16 \times 10^{-6}$ M), which is in accordance with the pSenLys biosensor response to this amino acid (Fig. 2b and Table 1). The other two basic amino acids, L-arginine and L-lysine, are 70 times and 200 times less tightly bound by LysG, respectively. Interestingly, low binding affinity interactions of LysG with these two ligands appear to be driven by enthalpy changes ($\Delta H$) of $-5.6$ kcal mol$^{-1}$ (L-lysine) and $-6.13$ kcal mol$^{-1}$ (L-arginine), with only minor entropic contributions. Conversely, the unfavorable enthalpic term of L-histidine binding ($\Delta H = 3.19$ kcal mol$^{-1}$) is outweighed by a major increase in entropy ($-T\Delta S = -9.73$ kcal mol$^{-1}$). While endothermic interactions are frequently coupled to the release of solvent molecules from the protein surface[22], blind docking analysis as well as extended MD simulations could not support this hypothesis. The binding mechanism appears to be only slightly affected by the A219L amino acid substitution as the determined parameters and the overall affinity of LysG-A219L to L-histidine were almost unchanged ($K_D = 20.7 \times 10^{-6}$ M). In contrast, this single amino

acid substitution had a strong impact on L-arginine- and L-lysine-binding properties. Whereas the binding affinity of LysG-A219L to L-arginine decreased by a factor of 10, the affinity of L-lysine to this engineered LysG variant was below the detection limit. These results are in accordance with the results obtained during the in vivo experiments with LysG-A219L as part of the L-lysine-insensitive pSenHis biosensor.

To gain additional insights into ligand–receptor interaction, extensive MD in the mircosecond time range was performed for LysG and LysG-A219L, with and without L-histidine, L-lysine, and L-arginine in the ligand-binding site, to reveal conformational changes that typically occur on the nanosecond timescale[23]. Two 1 μs simulations of LysG revealed two frequently occupied conformations with a reaction coordinate that can be exactly described by the distance between Cα atoms of residues 219 and 96, which are 6.4 Å apart in the effector-occupied (closed) RD in the LysG-Arg structure (Fig. 4a). Therefore, distances <10 Å were assumed to represent a closed conformation, while greater distances correspond to an open conformation. LysG was found in 30% of 2 μs in the open confirmation (Fig. 4d). In contrast, LysG-A219L was only found 3% of the time in the open conformation during 1 μs of MD. A second 1 μs MD of LysG-A219L, started from the open conformation, resulted in a similar distribution, as LysG-A219L rapidly transitioned back into the closed conformation. A hydrophobic interaction between residue 219 and the hydrophobic region A26, L99, A214, and L122 appears to stabilize the closed conformation. Due to its elongated spatial extent, A219L can form a hydrophobic connection between the two domains. If one considers the open conformation as the active form of the TR, it is possible to explain the increased background activity of the biosensor variant pSenLys compared to pSenHis in the absence of any ligand. The larger likelihood of LysG as part of pSenLys to assume the open conformation corresponds to higher background activity, as compared to LysG-A219L as part of pSenHis. Docking calculations suggested that L-histidine and L-lysine bind in the same binding pocket as L-arginine, and provided starting conformations for MD. Substitution of L-arginine for L-histidine in the ligand-binding site resulted in relative occupation of the open conformation in 17 and 24% of the time for LysG and LysG-A219L during 250 ns MD, respectively. This agrees well with the observed activities for both transcriptional biosensor variants in presence of L-histidine. The most remarkable effect was seen when L-lysine is substituted into the ligand-binding site. Here, LysG assumes over 80% of the time the open conformation, compared to <4% for LysG-A219L (Fig. 4d). Close investigation of LysG-A219L reveals that L-lysine forms three hydrogen bonds with 124, 125, and 97 when A219L enforced the closed conformation, resulting in a very tight packing that stabilizes the closed conformation (Fig. 4b). Conversely, LysG did not support the tight binding of L-lysine, but rather positioned the substrate like a wedge between the two domains, stabilizing the open conformation (Fig. 4c). Simulation of L-arginine in the ligand-binding site of both TRs yielded no transitions from closed to open conformation on the sub microsecond timescale when started from the crystal structure. Perturbation of the L-arginine orientation by 180-degree rotation resulted in an opening of the conformation, and a subsequent reorientation of L-arginine into the crystal structure orientation for both TRs. For the perturbed starting configuration also, no transition into the stable closed conformation was observed, suggesting that the energy barrier between both states is higher than for the TRs by themselves or in complex with L-histidine or L-lysine. This corresponds well with the successful crystallization of only the LysG–L-arginine complex and verifies the correct positioning of L-arginine. Further, the high energy barrier explains why fluorescence for both TRs in

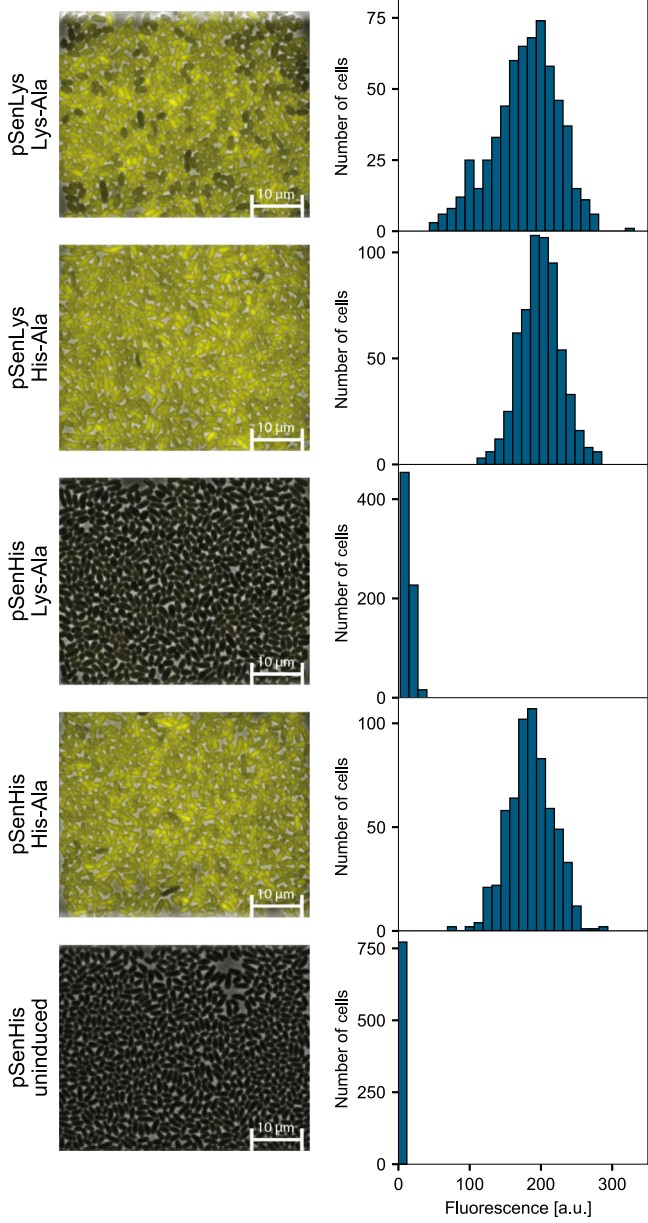

**Fig. 3 Microscale cultivations of *C. glutamicum* Δ*lysEG* carrying pSenLys or pSenHis.** At left: growth of cells in microfluidic chambers in the presence of L-His-L-Ala dipeptides, L-Lys-L-Ala dipeptides, or no dipeptides. At right: distribution of the corresponding fluorescence as measure of the biosensor response at the single-cell level across microcolonies.

presence of L-arginine was reduced compared to the other ligands. Conclusively, the closed conformation reduces the overall spatial extent of LysG, prohibiting the induction of gene expression, whereas the open conformation increases the protein size leading to LysG-mediated activation of gene expression. This explains the insensitivity of the biosensor pSenHis for L-lysine as LysG-A219L assumes the closed conformation in the presence of L-lysine, but responds to L-histidine just like the wild-type LysG protein.

**FACS-based high-throughput screening using pSenHis.** Finally, the engineered pSenHis biosensor was used to screen a culture of mutagenized *C. glutamicum* wild-type cells with the aim to isolate L-histidine-accumulating variants. For this purpose, a culture of the *C. glutamicum* ATCC 13032 wild type was chemically mutagenized by incubation with *N*-methyl-*N'*-nitro-*N*-nitroso-guanidine (MNNG). Of this culture, $10^7$ cells were screened using FACS, and 450 variants were isolated and individually cultivated for further characterization. HPLC analysis of culture supernatants revealed that 217 variants accumulated >0.1 mM L-histidine, whereas no L-histidine could be detected in supernatants of the *C. glutamicum* wild type. In addition, none of the isolated *C. glutamicum* variants accumulated L-lysine or L-arginine. Targeted sequencing of ten chromosomal genes involved in L-histidine synthesis (*hisA*, *hisB*, *hisC*, *hisD*, *hisE*, *hisF*, *hisG*, *hisH*, *hisI*, and *hisN*) of 25 isolated strain variants with the highest L-histidine accumulation in the supernatant ranging from 0.4 to 0.7 mM revealed that all variants bear mutations in the *hisG* gene encoding for the ATP-phosphoribosyltransferase (Table 2). HisG catalyzes the first committed step in the tightly controlled L-histidine pathway and is noncompetitively feedback-inhibited by elevated intracellular L-histidine concentrations[24,25]. Except for S143F, all identified mutations leading to amino acid substitutions in HisG are located in the allosteric binding site, in which a well ordered hydrogen bonding network interacts with L-histidine (Fig. 5b)[26]. Possibly, perturbation of this hydrogen bonding network induced by the identified amino acid substitutions weakens interaction with the pathway product L-histidine and keeps HisG in the active conformation.

Currently, additional rounds of FACS screening with the engineered pSenHis sensor are being conducted for identifying genomic targets contributing to L-histidine overproduction in *C. glutamicum*. These genomic hot spots will help to engineer *C. glutamicum* strains toward industrial-scale production of this amino acid, underlining the importance of biosensor-based screenings for rapid strain development.

## Discussion

Implementation of biosensor-based screening technologies can significantly shorten development times in the biotechnological industries, but they have also a huge impact on basic science as they help to uncover interrelations in the cellular metabolism and push forward the understanding of structure–function relationships in enzymes[6,18]. In cases where biosensor parameters, such as sensitivity or operational range do not fit the requirements of a certain application, these can be adapted by promoter engineering as it has been reported for a *cis,cis*-muconic acid biosensor in *Saccharomyces cerevisiae*[27]. In addition, the selectivity of the TR controlling reporter gene expression in response to the presence of the molecule(s) of interest can be expanded or shifted[12–16]. However, in this study, we showed that TRs can also be engineered toward a more focused ligand spectrum without changing other biosensor characteristics, which is of importance when a biosensor-based screening campaign yields only false-positive variants accumulating a different biosensor-detectable metabolite. Key to success was the FACS-based screening/counterscreening in the alternating presence of the desired ligand (L-histidine, positive screening) or undesired ligand (L-lysine, negative screening). A similar screening strategy has also been used to expand the ligand spectrum of the *E. coli* TRs LacI[15] and AraC[12–14] to different molecules. However, compared to the LacI studies where the TR was mutagenized by protein-wide single amino acid saturation mutagenesis and error-prone PCR, LysG was carefully redesigned by structure-guided and pairwise mutagenesis of selected residues in the ligand-binding site prior to FACS. This approach, allowing only for two amino acid substitutions per variant, enabled the isolation of LysG-A219L, bearing only a single amino acid substitution. As demonstrated by the performed ITC experiments and single-cell cultivations using microfluidics, this minimal modification of LysG ensured an

**Table 1 Thermodynamic binding parameters of LysG and LysG-A219L.**

| Protein | Ligand | $K_D$ (M)[a] | $\Delta G$ (kcal mol$^{-1}$)[a] | $\Delta H$ (kcal mol$^{-1}$)[a] | $-T\Delta S$ (kcal mol$^{-1}$)[a] |
|---|---|---|---|---|---|
| LysG | L-His | $(16 \pm 1.1) \times 10^{-6}$ | $-6.55 \pm 0.69$ | $3.19 \pm 0.47$ | $-9.73 \pm 0.5$ |
| LysG | L-Lys[b] | $(3.29 \pm 0.62) \times 10^{-3}$ | $-3.39 \pm 1.84$ | $-5.6 \pm 1.3$ | $2.2 \pm 1.3$ |
| LysG | L-Arg[b] | $(1.15 \pm 0.06) \times 10^{-3}$ | $-4.01 \pm 0.27$ | $-6.13 \pm 0.17$ | $2.11 \pm 0.21$ |
| LysG-A219L | L-His | $(20.7 \pm 2.64) \times 10^{-6}$ | $-6.39 \pm 0.52$ | $2.24 \pm 0.36$ | $-8.63 \pm 0.37$ |
| LysG-A219L | L-Lys[c] | n.a. | n.a. | n.a. | n.a. |
| LysG-A219L | L-Arg[b] | $(9.62 \pm 1.2) \times 10^{-3}$ | $-2.76 \pm 2.41$ | $-13.5 \pm 1.71$ | $10.8 \pm 1.7$ |

Binding parameters with regard to L-histidine-, L-lysine- and L-arginine binding as determined by isothermal titration calorimetry. Titrations were performed in phosphate buffer (50 mM NaH$_2$PO$_4$/Na$_2$HPO$_4$, 500 mM NaCl and 10% (v/v) glycerol), at pH 8.0 and 298 K.
[a]Errors reported as standard deviation (s.d.) from three independent experiments.
[b]Due to low binding affinities, released heat was fit using fixed stoichiometry of two ligand molecules binding to one LysG-tetramer.
[c]Below detection limit (dissociation constant ($K_d$): 10$^{-9}$ M).

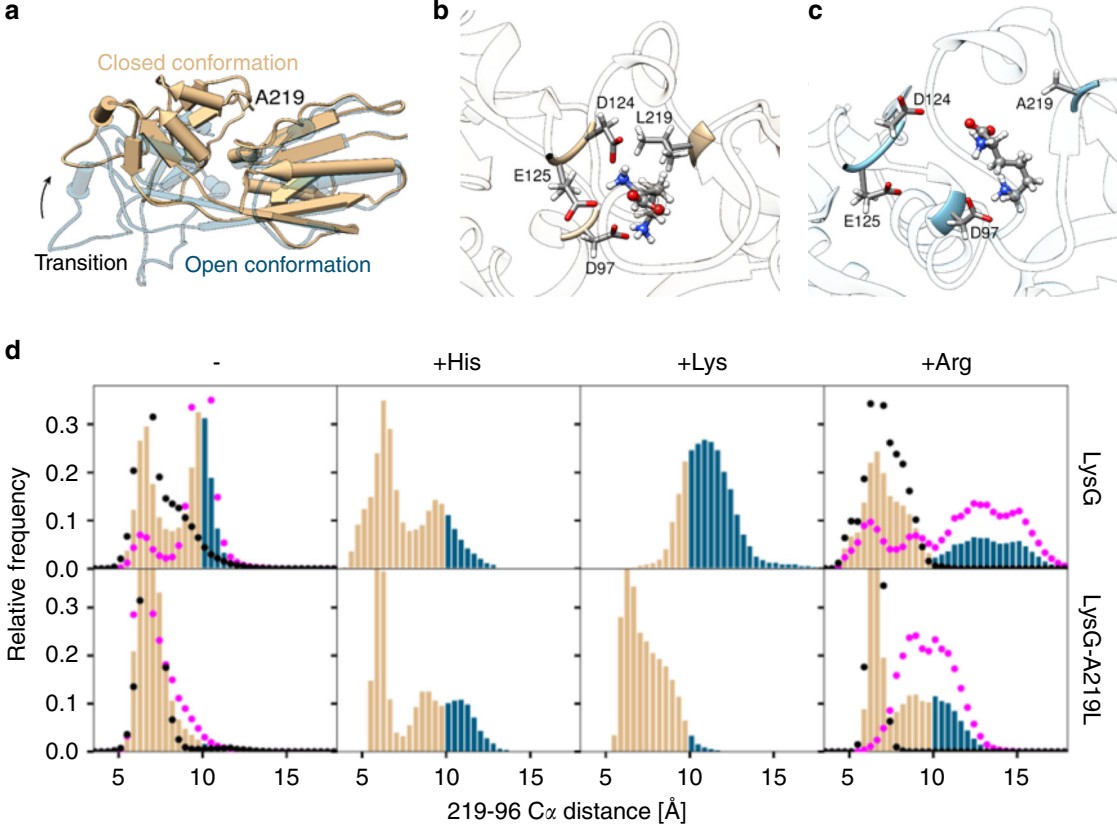

**Fig. 4 Distribution of open and closed conformations of LysG and LysG-A219L binding domains. a** The 2 µs simulation of LysG and 2 µs simulation of LysG-A219L. Both samples have an open (blue) and closed (tan) conformation of the ligand-binding domain, where openness is defined by distance >10 Å between atoms 96 Cα and 219 Cα. It is proposed that the open conformation corresponds to an *eyfp* fluorescence promoting sensor. **b** Three predominant hydrogen bridges between L-lysine and the L219 linker stabilize the closed conformation in LysG-A219L. **c** Wedge function of LysG that forces the open conformation of the regulator to remain stable. **d** Distributions of open (blue) and closed (tan) conformations as sampled in simulations for TRs LysG (two times 1 µs—black (from crystal conformation) and magenta (from open start conformation) dots) and LysG-A219L (two times 1 µs—magenta and black dots), and for complexes of both sensors with ligands L-lysine and L-histidine in the respective binding site (four times 250 ns). For L-arginine, joint distance distributions from two starting configurations are shown (four times 250 ns—black (from crystal conformation) and magenta (from rotated ligand conformation) dots).

unaltered biosensor response to the presence of L-histidine and L-arginine, whereas no biosensor response to L-lysine could be detected. The original biosensor pSenLys was used in a FACS campaign to identify mutagenic hot spots contributing to L-lysine biosynthesis in chemically mutagenized *C. glutamicum* cultures[6]. Of 270 variants isolated by FACS and analyzed in detail, no variant accumulated L-histidine or L-arginine, and all subsequent attempts to isolate L-histidine-producing variants failed. In contrast, when using the engineered biosensor pSenHis in an ultra-high-throughput screening campaign as presented here, only L-histidine-accumulating variants were isolated.

Extensive MD simulations of LysG and LysG-A219L suggest that observed biosensor activities of pSenLys and pSenHis depend on the conformation of the regulator, as also frequently observed in regulators of G protein signaling (RGS)[25]. The distributions of the open and closed conformations correspond to measured fluorescence, where an open conformation suggests an activated state. This bimodal distribution is strongly influenced by the

**Table 2 Amino acid substitutions in HisG and determined L-histidine concentrations.**

| Amino acid substitution(s) in HisG | L-His titer (mM) |
| --- | --- |
| Wild type | 0 |
| S143F | 0.67 ± 0.21 |
| S143F + G233D | 0.43 ± 0.1 |
| D213N | 0.68 ± 0.19 |
| G230S | 0.57 ± 0.09 |
| G230D | 0.57 ± 0.23 |
| S232P | 0.46 ± 0.08 |
| G233D | 0.62 ± 0.14 |
| T235M | 0.47 ± 0.17 |

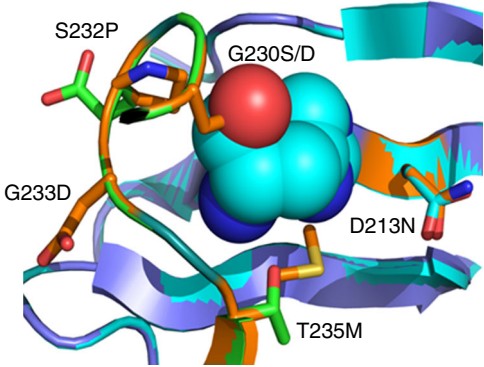

**Fig. 5 Amino acid substitutions in the ATP-phosphoribosyltransferase (HisG).** Location of the amino acid substitutions in the allosteric L-histidine binding site in HisG of *C. glutamicum*. The structure model for HisG of *C. glutamicum* was built based on the HisG crystal structure of the closely related *Mycobacterium tuberculosis*[26] (PDB code: 1NH8) using SWISS-MODEL[48].

respective substrate. Whereas allosteric inhibition has been revealed by MD for other regulators previously[23,28–30], this study also explains why the single substitution of a small alanine for a larger leucine residue narrows the substrate spectrum. While allosteric transitions frequently occur on longer timescales, the observed transitions are impacted by ligand binding at the hinge region of the two-lobe regulator domain. It should be noted that the simulation does not describe the allosteric conformational change. The simulation only considers the conformational motion of the two-lobe regulator domain and how it is influenced by different ligands. The resulting allosteric effect is the activation of the transcription factor in the DNA-binding domain, which happens further away as a consequence of this domain motion, and is not studied here. This substitution enables a hydrophobic interaction between A219L and a hydrophobic patch on the opposite domain that functions like a latch. This latch stabilizes the closed conformation and causes bound L-lysine to find a snug conformation inside the ligand-binding site, similar to inhibition of an RGS by the selective inhibitor CCG$_{-50014}$ (ref. [28]). Furthermore, L-lysine is stabilized by the formation of three hydrogen bonds. In contrast to this, L-histidine does not stabilize the closed conformation. It rather wedges itself between A219L and the opposite domain to stabilize the open conformation of LysG and LysG-A219L. The steric extent of L-histidine and the more restricted flexibility compared to L-lysine explain why a single amino acid substitution can decrease the ligand spectrum of this TR, allowing for the construction of a biosensor with a focused ligand spectrum. Further analysis of the L-arginine interactions with LysG might suggest additional substitutions that will enable the construction of a pSenArg sensor.

No other TR as part of a biosensor has been studied in such structural detail, even though the LTTR LysG is a member of the largest known family of prokaryotic TRs comprising >40,000 proteins plus many more functionally orthologous proteins identified in archaea and eukaryotic organisms (Interpro Entry: IPR000847)[20,31]. Since they can act either as activators or repressors of single or operonic genes and are known to accept a broad range of chemically diverse ligands, it is no surprise that many different LTTR-based biosensors have been constructed[16,27,32]. The strategy for altering ligand-binding properties of LTTRs along with a detailed understanding of the ligand-binding mode of these TRs presented in this study should spur interest in developing custom-made LTTR-based biosensors. Possible applications go beyond much-needed ultra-high-throughput screens for optimizing biocatalysts to produce chemicals as shown in the context of this study. Additional applications include using engineered LTTRs for designing ligand-inducible genetic circuits to reprogram the cellular metabolism or for developing growth-coupled selection methods for bio-based production of chemicals.

## Methods

**Bacterial strains, plasmids, media, and growth conditions.** All bacterial strains and plasmids used in this study, and their relevant characteristics are listed in Supplementary Table 2. *C. glutamicum* was routinely cultivated aerobically at 30 °C in brain heart infusion (BHI) medium (Difco Laboratories, Detroit, USA) or defined CGXII medium with 4% glucose, as sole carbon and energy source[33]. *E. coli* DH5α used for plasmid constructions was cultivated in LB medium[34] at 37 °C, *E. coli* BL21(DE3) used for heterologous gene expression was cultivated in 2× YT medium. Where appropriate, kanamycin (50 μg mL$^{-1}$ for *E. coli* or 25 μg mL$^{-1}$ for *C. glutamicum*) or spectinomycin (100 μg mL$^{-1}$ for *E. coli* and *C. glutamicum*) was added to the medium. Bacterial growth was followed by measuring the optical density at 600 nm (OD$_{600}$). Precultures of *C. glutamicum* were grown for 6–8 h in test tubes with 5 mL BHI medium on a rotary shaker at 170 r.p.m. Subsequently, 50 mL defined CGXII medium with 4% glucose in 500 mL baffled Erlenmeyer flasks were inoculated with cells from the preculture and cultivated on a rotary shaker at 130 r.p.m.

**Construction of plasmids and strains.** Standard protocols of molecular cloning, such as PCR, DNA restriction, and ligation[35] were carried out for recombinant DNA work. All oligonucleotides used in this study were obtained from Eurofins Genomics (Ebersberg, Germany) and are listed in Supplementary Table 3. Techniques specific for *C. glutamicum*, e.g. electroporation for transformation of strains, were performed according to standard protocols[36]. All enzymes were obtained from ThermoScientific (Schwerte, Germany), all dipeptides were purchased from Bachem (Bubendorf, Switzerland). All constructed plasmids, isolated regulator variants, and genome-encoded *his*-genes were sequenced at Eurofins Genomics (Ebersberg, Germany).

**Expression and purification of LysG.** A single colony of *E. coli* BL21(DE3) pET28b(+)-*lysG* was used to inoculate 5 mL 2× YT medium containing 50 μg mL$^{-1}$ kanamycin. This preculture was used to inoculate 500 mL 2× YT medium containing 50 μg mL$^{-1}$ kanamycin, which was subsequently incubated for 3 h at 37 °C. IPTG was added to a final concentration of 1 mM, and the cultivation continued for 16 h at 25 °C. Cells were harvested, washed in buffer (50 mM Sorensen's phosphate buffer, pH 8, 300 mM NaCl, 10 mM imidazole, and 0.3 mM DTT), and resuspended in the same buffer prior to cell disruption by sonication (Branson Ultrasonics Corporation, Danbury, CT, USA; eight cycles of 30 s, 4 °C, duty cycle 34, output control 8). Crude extracts derived after centrifugation (230,000 × *g*, 4 °C, 1 h) were mixed with 5 mL nickel-nitrilotriacetic acid (Ni-NTA) beads (Qiagen, Hilden, Germany) and incubated for 1 h at 4 °C. This mixture was applied to gravity-flow columns, equilibrated in 50 mM Sorensen's phosphate buffer, pH 8, 300 mM NaCl, and 0.3 mM DTT. The columns were washed with four volumes of Sorensen's phosphate buffer and the protein was eluted with 50 mM Sorensen's phosphate buffer, pH 8, 300 mM NaCl, 250 mM imidazole, and 0.3 mM DTT. LysG-containing fractions were determined by SDS–PAGE, pooled and dialyzed overnight in 20 mM Tris-HCl pH 8, 500 mM NaCl, and 0.1 mM DTT. The protein was concentrated to 10 mg mL$^{-1}$ using Amicon Ultra-4 30 K centrifugal filters (Millipore Corporation, Billerica, MA, USA). Size-exclusion chromatography was performed on an Äkta-P900 System (GE Healthcare, Chicago, IL, USA) equipped with a Superdex 200 10/300 GL column and a UPC-900 detection unit. The column was buffered in 50 mM Sorensen's phosphate buffer, pH 8, 150 mM NaCl, and calibrated with the Gel Filtration Markers Kit (Sigma-Aldrich, St. Louis, Mo, USA) as recommended by the manufacturer. Data were analyzed with the Unicom 5.01 software (GE Healthcare, Chicago, IL, USA).

**Crystallization of LysG, structure determination, and refinement.** Initial crystals for LysG were obtained by nanodrop crystallization (100 nl + 100 nl) using a Phoenix crystallization robot (Art Robbins Instruments, Sunnyvale, CA, USA) and Nextal crystallization screens (Qiagen, Hilden, Germany). Ligand-free LysG was crystallized in hanging drop plates by mixing 1 μl reservoir (17.5% PEG4000, 0.1 M ammonium sulfate, and 0.1 M Tris-HCl pH 8.0) and 1 μl of LysG (4.5 mg ml⁻¹) at 291 K. Large crystals were transferred to reservoir solution supplied with 20% glycerol for cryoprotection. A diffraction data set was collected at ESRF beamline ID29 (European Synchrotron Radiation Facility, Grenoble, France) at a wavelength of 1.044 Å and 100 K. Cocrystallization of LysG with L-arginine was achieved by preincubation (15 min, 291 K) of 10 mM L-arginine with 0.34 mM protein in solution. A LysG + Arg crystal was harvested directly from the MbClass II C2 screen (0.1 M NaCl, 0.1 M Tris pH 8.5, and 30% (v/v) PEG400) and cryoprotected by adding 15% sucrose. Diffraction data were collected at ESRF beamline BM30A at wavelength 0.978 Å and 100 K. XDS[37] and XSCALE[37] were used for data processing and scaling. For the calculation of $R_{free}$, 5% of the data were randomly assigned. The crystal structure of ArgP from *Mycobacterium tuberculosis* (PDB ID: 3ISP)[38] was used as search model for molecular replacement using Phaser[39]. The structure was manually improved in Coot[40] and automatically refined in Phenix[39]. Torsion angle non crystallographic symmetry restraints were used throughout. The N-terminal expression tag and residues 192–196 could not be modeled due to missing density. The structure was refined to $R$ and $R_{free}$ values of 19.47% and 22.74%, respectively. A total of 96.28% of the residues are in the favored region of the Ramachandran plot, 3.37% and 0.35% are in the allowed and disallowed regions, respectively. The generated structure of LysG was used as search model for LysG + Arg. A similar refinement protocol was applied. Due to the lower resolution, additional geometry restraints against the LysG model were used throughout. The structure was refined to $R$ and $R_{free}$ values of 20.65% and 25.08%, respectively. A total of 97.01% of the residues are in the favored regions of the Ramachandran plot, 2.99% are in the allowed regions, no residues are in disallowed regions. The expression tag and residues 194–198 were not included in the final model due to disorder.

**Generation of semi-rational *lysG* libraries.** The LysG structure without ligand was used as target molecule to dock L-histidine or L-lysine by the SwissDock webserver[41]. Residues N95, D97, R143, D124, E125, H161, F189, G190, P191, D193, A219, and F222 were selected for pairwise site-saturation mutagenesis according to the QuikChange protocol (Agilent Technologies, Santa Barbara, CA, USA). For this purpose, the template plasmid pEKEx3-*lysG* and mutagenic oligonucleotides containing NNS codons were used to limit the number of genetically unique variants in each library to 1024, while still allowing for the discovery of synergistic mutations (Supplementary Table 3). Subsequently, the mutagenized *lysG* variants were cloned into the *pSen* sensor plasmid carrying the open reading frame of the *eyfp* reporter gene under control of the *lysE* promoter, which in turn is activated by the encoded LysG variants.

**FACS-based screening of biosensor variants.** *C. glutamicum* Δ*lysEG* was transformed separately with all six *lysG* mutant libraries present on engineered pSenLys. All mutant libraries were combined and cultivated in defined CGXII medium for 1 day at 30 °C. For maintenance of the pSenLys plasmid and its derivatives, 25 mg L⁻¹ kanamycin were added. Subsequently, the precultures were diluted to an OD₆₀₀ of 0.5 in fresh CGXII medium containing 3 mM L-His-L-Ala dipeptide. Equally treated precultures of *C. glutamicum* Δ*lysEG* carrying the biosensor plasmid pSenLys with the wild-type *lysG* gene, served as either positive control (+3 mM L-His-L-Ala) or negative control (no dipeptide supplementation) during all FACS-based screening and counterscreening experiments. For this purpose, all cultures were grown for 7 h at 30 °C and diluted to an OD₆₀₀ < 0.1 in FACSFlow (BD) prior to FACS. Subsequently, cells were subjected to single-cell autofluorescence analysis using a FACSAria II (BD Biosciences, Franklin Lakes, NJ, USA) equipped with a 70 μm nozzle and run with a sheath pressure of 70 p.s.i. A 488 nm blue solid laser was used for excitation. Forward-scatter characteristics (FSC) were recorded as small-angle scatter and side-scatter characteristics (SSC) were recorded as orthogonal scatter of the 488 nm laser. A 502 nm long-pass and 530/30 nm band-pass filter combination enabled EYFP fluorescence detection. Prior to data acquisition, debris was excluded from the analysis by electronic gating in the FSC-H against SSC-H plot. Using the fluorescence output of *C. glutamicum* Δ*lysEG* pSenLys induced with 3 mM L-His-L-Ala dipeptide (positive control), 200,000 cells characterized by similar or higher fluorescence were sorted into 5 mL reaction tubes (Eppendorf AG, Hamburg, Germany), pre-filled with 3 mL fresh CGXII medium (positive sorting). After cultivation for 2 days at 30 °C, these cultures were used to inoculate fresh CGXII medium with 3 mM L-Lys-L-Ala dipeptide. Following a second cultivation for 7 h at 30 °C, cells were diluted in FACSFlow for single-cell autofluorescence measurements as outlined above. In contrast to the previous positive sorting, the fluorescence parameters of *C. glutamicum* Δ*lysEG* (pSenLys) grown in absence of dipeptide (negative control) was used to isolate cells without or with reduced fluorescence into fresh CGXII medium (negative sorting). The resulting cultures were subsequently subjected to an additional round of positive and negative sorting as described above.

*C. glutamicum* cells isolated in the second negative FACS sorting step were collected in defined CGXII medium supplemented with 3 mM L-His-L-Ala

dipeptide and, after growth for 7 h in 30 °C, single cells characterized by a higher fluorescence compared to the positive control were collected in fresh CGXII medium using FACS to finally obtain biosensor variants insensitive to L-lysine. FACSDiva 7.0.1 (BD Biosciences, San Jose, USA) was used to control the FACS device and to perform data analysis. FlowJo for Windows 10.4.2 (FlowJo, LLC, Ashland, OR, USA) and Prism 7.04 (GraphPad Software, San Diego, CA, USA) were used to produce high-resolution graphics of FACS data.

**Characterization of isolated LysG variants.** Initially, biosensor plasmids of 14 clones obtained during the FACS-based screening and counterscreening experiments were isolated and retransformed into *C. glutamicum* Δ*lysEG*. Then, these clones were cultivated in CGXII medium containing either L-His-L-Ala, L-Lys-L-Ala, or L-Ala-L-Ala dipeptides (3 mM) at 30 °C and 900 r.p.m. using a BioLector cultivation system (m2p-laboratories GmbH, Baesweiler, Germany), which was also used to simultaneously follow fluorescence formation of individual cultivations. Formation of biomass was recorded as the backscatter light intensity (wavelength 620 nm; gain factor 25). EYFP fluorescence was determined as fluorescence emission at 532 nm (gain factor: 30) after excitation at 510 nm. The specific fluorescence was calculated as 532 nm fluorescence per 620 nm backscatter using Biolection software version 2.2.0.6 (m2p-laboratories GmbH, Baesweiler, Germany).

The L-lysine-insensitive biosensor variant pSenHis carrying LysG-A219L was characterized in a *C. glutamicum* Δ*lysEG* strain background for its fluorescence in response to the presence of L-His-L-Ala, L-Lys-L-Ala, or L-Ala-L-Ala dipeptides at concentrations ranging from 0 to 5 mM, using the BioLector system with the same settings as described above. For this purpose, cells from CGXII precultures were diluted in fresh CGXII medium to an OD₆₀₀ of 1. Following backscatter (wavelength 620 nm; gain factor 25) and biosensor fluorescence at 532 nm (gain factor 30, excitation at 510 nm), these main cultures were incubated at 30 °C, 900 r.p.m. for 2 h before concentrated dipeptide stocks were added to each well.

**Single-cell cultivations.** The polydimethylsiloxane-based microfluidic single-cell cultivation system utilized in the present study allowed for growth and metabolic studies of ~500–1000 microbial cells at defined and precise environmental conditions. Cells were cultivated within 50 × 50 μm monolayer growth chambers (~1 μm in height), which are connected to 80 μm wide and 10 μm deep main channels to keep the cell numbers in each growth chamber nearly constant while supplying fresh medium[42].

For all live-cell imaging experiments, cell suspensions of exponentially growing *C. glutamicum* cells (OD₆₀₀ of 0.3–0.5) in defined CGXII medium were used to inoculate the microfluidic chips. Before starting an experiment, appropriate growth chambers were selected manually. After a preliminary growth phase in defined CGXII medium with 4% glucose, a biosensor response was triggered by switching to CGXII medium containing 3 mM of the corresponding dipeptide. Microscopy images were taken using an inverted microscope (Nikon TI-Eclipse, Nikon Instruments, Tokyo, Japan) equipped with a 100× oil immersion objective (CFI Plan Apo Lambda DM 100×, NA 1.45, Nikon Instruments, Tokyo, Japan) and a temperature incubator (PeCon GmbH, Erbach, Germany). Phase-contrast and fluorescence time-lapse images were recorded every 10 min using an Andor Luca R DL 604 CCD camera (Andor Technology Ltd, Belfast, Northern Ireland, UK), with appropriate filter sets. The cell areas and fluorescence values obtained from time-lapse movies were analyzed semi-automatically using NIS-Elements Microscope imaging software (Nikon Instruments, Tokyo, Japan) in combination with manual inspection and correction.

**Purification of LysG and LysG-A219L for ITC experiments.** Precultures of *E. coli* BL21(DE3) variants engineered for the heterologous expression of *lysG* or *lysG*-A219L were cultivated overnight in 2× YT medium containing 50 μg mL⁻¹ kanamycin at 37 °C, 150 r.p.m. The following day, 3 mL of these precultures were used to inoculate 300 mL 2 × YT medium containing 50 μg mL⁻¹ kanamycin. After cultivation for 3 h at 37 °C, the incubation temperature was reduced to 25 °C and heterologous gene expression was induced by supplementation of 1 mM IPTG. Cells were harvested by centrifugation (6350 × *g*, 60 min, 4 °C) after 16 h and the cell pellets were washed with buffer A (50 mM Na₂HPO₄/NaH₂PO₄, 500 mM NaCl, pH 8.0). Optionally, cell pellets were stored at −80 °C. For cell lysis, cell pellets were resuspended in 12 mL buffer A with 10% glycerol and sonicated on ice using an ultrasonic cell disruptor (Branson Ultrasonics Corporation, Danbury, CT, USA), eight sonication cycles of 30 s, 4 °C, duty cycle 34 and output control 8). After two centrifugation steps (7150 × *g*, 30 min, 4 °C followed by 230,000 × *g*, 1 h, 4 °C) the cleared lysates were loaded on gravity-flow columns filled with 1 mL Ni-NTA affinity agarose (Qiagen, Hilden, Germany) previously equilibrated with buffer A, 10% glycerol, and 20 mM imidazole. Subsequently, the columns were washed twice with 10 mL buffer A, 10% glycerol, 20 mM imidazole, and the proteins were eluted using 10 mL buffer A, 10% glycerol, and 250 mM imidazole. After desalting using PD-10 columns (GE Healthcare, Chicago, IL, USA), the protein was used immediately for further steps. Protein concentrations were determined using a Nano-Drop ND-1000 spectrophotometer (Thermo Fisher Scientific, Waltham, MA, USA) and ranged from 25 mg L⁻¹ to 100 mg L⁻¹.

**Isothermal titration calorimetry**. The protein was diluted or concentrated to 50 μM using centrifugal filters (Amicon Ultra-15, MCWO 10 kD, Millipore, Merck, Darmstadt, Germany) and dialyzed overnight against a 500-fold excess of dialysis buffer (50 mM Na$_2$HPO$_4$/NaH$_2$PO$_4$, 500 mM NaCl, 10% (v/v) glycerol, pH 8.0), using a membrane tube with a 6–8 kD MWCO (SpectrumLabs, Piraeus, Greece). Prior to experiments, concentrated amino acid stock solutions were prepared and diluted to the desired concentrations using the dialysis buffer. ITC measurements were performed with a MicroCal PEAQ-ITC (Malvern Panalytical, Kassel, Germany) operated at 25 °C. After rinsing with dialysis buffer, the measuring cell was filled with 300 μL protein solution and the syringe was filled with 75 μL amino acid solution. Each ITC run was started with an initial 0.4 μL injection followed by 12 3 μL injections. All experiments were performed in triplicate and all data were analyzed using the MicroCal ITC analysis software (Malvern Panalytical, Kassel, Germany).

**Computational modeling**. Models were prepared from crystal structures 6XTU (LysG) and 6XTV (LysG + Arg). For simulation of LysG residues 85–290 from chain A were extracted from 6XTU (the missing loop between residues 191 and 197 was modeled according to weak density signal and removed prior to submission of structure to the PDB). A structure for LysG-A219L was obtained by mutating A219L in the LysG structure, using CHIMERA[43] and the Dunbrack rotamer library[44]. For the second LysG-A219L simulation an open conformation sampled in the first LysG trajectory was chosen and mutated to A219L. For simulation of complexes, residues 89–290 were extracted from 6XTV chain B and again mutated at position 219 from A to L with CHIMERA to yield the receptor structures. The residues L-histidine, L-lysine, and L-arginine were parametrized in the General Amber Force Field with ambertools[45].

The position of L-arginine was taken from 6XTV chain B and manually rotated by 180 degrees around its center of mass for each receptor structure to yield four start conformations. The positions of L-histidine and L-lysine were obtained by docking runs performed using flexible ligand docking in AutoDock Vina 1.1.2 (ref. [21]) for the two receptor structures. In addition, the L-arginine position was calculated and compared to the experimental conformation to verify the method. The docking search space was set over the entire protein to allow for identification of alternative binding sites. To increase the exhaustiveness of the search over such a large box, the exhaustiveness parameter was increased to 80 and ten separate runs were performed per complex. The top scoring pose of each AutoDock Vina run fell within 1.3 Å center of mass distance from the crystal position of L-arginine into the same binding pocket. The best scoring pose from each complex was selected to define the start conformations of the L-lysine and L-histidine complex simulations.

The systems were prepared in a cubical box with 10 Å padding between the protein and closest wall. The box was solvated with water coordinates from spc216. The system was neutralized with Na+ and Cl− ions. The systems were minimized using steepest descent algorithm with a step size of 0.01, a convergence criterion of 1000 kJ mol$^{-1}$ nm$^{-1}$ and a maximum number of 50,000 steps. For the minimization, system setup and subsequent MD, the program GROMACS[46] was used with the AMBER99SB-ILDN[47] force field and tip3p water model. Following minimization, an NVT (constant number of particles, volume, and temperature) equilibration over 100 ps with cubic periodic boundary conditions, leapfrog Verlet integrator with time steps of 2 fs, holonomic constraints with LINCS iteration of 1 and order 4, and restrained hydrogen bonds was conducted. The cutoffs for short range electrostatics and van der Waals were set to 1 nm. Particle Mesh Ewald was used for long range electrostatics with PME order 4 and a grid spacing of 1.6 Å. Temperature coupling used the velocity rescaling (v-rescale) thermostat at 300 K and coupled the protein to the solvent with a time constant of 0.1 ps. Next, a 100 ps NPT (constant number of particles, pressure, and temperature) simulation followed with the same parameters plus the addition of an isotropic Parrinello–Rahman pressure coupling with time constant of 2 ps at 1 bar pressure, given a compressibility of $4.5e^{-5}$ bar$^{-1}$ for water. Following equilibrations, productive MD runs were started with the same parameters as NPT. For LysG and LysG-A219L two 1 μs simulations were calculated. For each of the eight complex systems 250 ns simulations were conducted, resulting in a total of 6 μs simulation. The docking poses, parametrizations, and run input files of productive MD are made available at https://simtk.org/projects/lysg.

**pSenHis-based FACS screening for L-histidine producers**. *C. glutamicum* ATCC 13032 carrying pSenHis was grown in 5 mL BHI complex medium (Difco Laboratories Inc., Detroit, MI, USA) containing 25 μg mL$^{-1}$ kanamycin to an OD of 5 (exponential growth phase). Whole-cell mutagenesis was performed by the addition of MNNG dissolved in dimethyl sulfoxide to a final concentration of 0.1 mg mL$^{-1}$ and incubation for 15 min at 30 °C. Subsequently, treated cells were washed twice with 45 mL NaCl, 0.9% (w/v), resuspended in 10 mL BHI, and regenerated for 3 h at 30 °C and 180 r.p.m. Afterward, the mutagenized cells were stored at −30 °C as cryostocks (BHI, 40% glycerol (w/v)). For FACS screening, the mutant library containing $4.2 \times 10^8$ viable cells mL$^{-1}$ was diluted 1:100 in 20 mL defined CGXII medium. After 2 h of cultivation, $10^7$ cells were analyzed by FACS as described above and 450 cells spotted on Petri dishes containing agar with defined CGXII medium. Colonies grown after 48 h at 30 °C were further analyzed.

**Amino acid quantification**. The three basic amino acids were quantified as their *o*-phthaldialdehyde derivatives via high-pressure liquid chromatography using an uHPLC 1290 Infinity system (Agilent Technologies, Santa Clara, CA, USA) equipped with a Zorbax Eclipse AAA C18 3.5 micron 4.6 × 75 mm and a fluorescence detector. As eluent, a gradient of 0.01 M Na-borate buffer pH 8.2 with increasing concentrations of methanol was used, and detection of the fluorescent isoindole derivatives was performed using an excitation wavelength of 230 nm and an emission wavelength of 450 nm.

**Reporting summary**. Further information on research design is available in the Nature Research Reporting Summary linked to this article.

## Data availability

Data supporting the findings of this manuscript are available from the corresponding author upon reasonable request. A reporting summary for this article is available as a Supplementary Information. Coordinates and structure factor amplitudes for the apo structure of LysG and LysG complexed with l-arginine were deposited at the Protein Data Bank (PDB) under accession codes PDB 6XTU and PDB 6XTV, respectively and are publicly available. Furthermore, the docking poses, parametrizations, and run input files of productive MD are made available at https://simtk.org/projects/lysg. Source data are provided with this paper.

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

## Acknowledgements

This project has received funding from the European Research Council (ERC) under the European Union's Horizon 2020 research and innovation program (grant agreement no 638718), and the German Federal Ministry of Education and Research (BMBF; FlexFit, grant no 0315589 A). We thank the beamline staff at beamlines ID29 and BM30A at the European Synchrotron Radiation Facility (ESRF), Grenoble, France, for the support during data collection and would like to acknowledge the help of Georg Schaumann, Michael Bott, and Lothar Eggeling in the initial phase of the project.

## Author contributions

J.M., D.D.C., and M.S. conceived and designed the study. E.H., F.S., and C.S. crystallized LysG and determined the structure, M.S. and F.T. generated LysG libraries by pairwise site-directed mutagenesis and isolated L-lysine-insensitive biosensor variants by FACS, S.S. and K.K. performed molecular cloning, heterologous gene expression, and protein purification. H.L.v.B. and S.S. characterized pSenHis and performed ITC measurements, K.U.C. analyzed and interpreted all ITC data, A.G. performed and analyzed all single-cell cultivations, D.D.C., C.J.M., and G.F.S. performed all computational modeling experiments, P.T.B. and M.S. performed chemical mutagenesis and FACS-based screenings, and characterized isolated L-histidine overproducing strain variants, and J.M. and D.D.C. wrote the manuscript with input from all authors.

## Funding

## Competing interests

The authors declare no competing interests.
