## [Peer Review File · Nature Communications]

Reviewer #1 (Remarks to the Author):

The authors sought a variant of LysG having reduced sensitivity toward lysine, in hopes of developing a biosensor that would enable high-throughput screening for mutants of *C. glutamicum* producing His. To do this, a crystal structure of LysG was first determined. Various combinations of residues potentially important to ligand binding were randomized and the libraries were screened by FACS to isolate L-lysine-insensitive variants that still respond to L-histidine. "pSenHis" was ultimately isolated and studied by ITC and MD. Finally, the sensor was used to isolate a variety of hisG mutant *C. glutamicum* strains that overproduced His.

Overall this is a well-written and thorough study that addresses the challenge of TF-based biosensor specificity. The experiments and described in sufficient detail, and statistical analysis is sufficient. My only concerns / questions relate to the ITC and MD results: We know that LysG responds well to lysine (essentially the same response as to his at low concentrations) – so why is the measured K_d so different? How can this be rationalized? Then, the MD results suggest that lysine acts to inhibit gene activation. Shouldn't that binding also be reflected in the ITC data? Why or why not? Finally, and related, results for the response of pSenHis in the presence of both his and lysine should be presented. Or some experimental analysis of LysG-A219L in the presence of both ligands. Is there clear competitive inhibition?

Reviewer #2 (Remarks to the Author):

This paper has shown a semi-rational engineering of a transcriptional regulator protein, LysG, by high-throughput screening of the site-saturated mutants based on the LysG and LysG-Arg high resolution structures. It demonstrates that the mutant A129L could significantly reduce the LysG affinity to L-lysine, which is novel and helps to screen L-histidine-producing strains efficiently.

The findings in this paper are useful for other researchers to continue the rational engineering of LysG for different needs in the amino acid biosensor. It encourages other researchers to spend time to obtain the high-resolution structures to assist the high-throughput screening of the target.

The authors obtained LysG-Arg structure but mainly showed the validation of pSenHis rather than pSenArg, which decreases the significance of detailed analysis and tight correlation between the structure and function. To prove that the binding site of His could be same with Arg in LysG, several different data are used to indirectly demonstrate it. However, it's still possible that the mutation site A129L influences other adjacent structures related to His, which behaves differently or even binds in a different site compared with Arg in LysG. And the result, in this paper, that none of *C. glutamicum* variants can accumulate L-Arg could be one evidence. It means that the mutations based on the analysis of LysG-Arg structure did not help to screen out cell variants for L-Arg accumulation.

Regarding to the positioning of Arg in the electron density map, it's not persuasive with the open-closed conformation analysis by MD. If the researcher can explain from other aspects about why Arg is not positioned 180° rotated, it will be better. D124 and E125 look to be more competitive than D193 to interact with the positively charged guanidino group of Arg.

The whole demonstration could be better if the authors could address above questions.

Reviewer #3 (Remarks to the Author):

I support the publication of this paper--with some modifications. It describes the semi-rational engineering of a desired specificity change of a protein and the utilization of this changed specificity to isolate a mutant bacteria overproducing the ligand of the engineered protein. The approach and

findings should be of interest to a wide audience.

My comments below are of two types. The first concern relatively minor changes that improve readability, understanding, or accuracy, and the second concern removing the docking and molecular dynamics components of the paper which I find to be unjustified and unsatisfactory.

First type of comment:

I. 74 In my experience, the vast majority of transcriptional regulators can be characterized as having a narrow specificity, not broad. The authors' characterization of known transcriptional regulators simply could to be changed to state that in a good number of cases, the ligand specificity of a transcriptional regulatory protein needs to be altered in order to permit its efficient utilization in a sensor system.

I. 87 The intended meaning is more clearly communicated with the omission of "large".

I97-100 It would really help the reader to state that the transcriptional regulator stimulates activity of a promoter that drives the synthesis of a fluorescent protein, which allows fluorescence activated cell sorting of individual cells. Also, at its first appearance in the paper, the abbreviation eyfp needs to be spelled out (I 138?).

I. 109-110 It sounds contradictory to state that the crystal structure of LysG had to be solved because there were no suitable homologs, and then to use molecular replacement to solve the structure. Perhaps the authors could tell us retrospectively whether it really was necessary to solve the crystal structure.

I. 260 Please provide the detection limit.

I. 359 Wasn't the redesign of AraC also structure-based?

I.386 But plenty of transcription factors have been studied in far more detail and depth that was done here for LysG. The depth of study and the coupling to a promoter and fluorescent protein are two independent operations.

In this same paragraph, one can't help wondering why the structure of LysG wasn't predicted using David Baker's approach of coupling the sequences of a large number of homologs to a prediction by Rosetta.

I believe it would enhance the paper to describe more carefully and completely how it is first of all, that LysG can respond to lysine, arginine, and histidine, and then why the mutant can no longer respond to lysine while still responding to the other two amino acids. The authors made a brief stab at part of this, but it was too hard for me to follow.

And now, the arguments for removal of all mention of docking and molecular dynamics.

With respect to docking. First note the authors' statement at the end of the legend of Fig. S 3. "We find that the docking tools do not agree very well amongst each other, and none of the findings provide a conclusive argument for alternative binding poses for L-histidine." Such a statement invalidates the statement made elsewhere in the paper relating to the number of water molecules released from the protein upon one ligand's binding. The precise use of the docking programs is not described and it does not appear that others in the field could exactly reproduce what the authors claim. Also, if such docking programs are effective, I would expect them to be used to find and improve potential drugs against medically important proteins. Are they? Finally, the paper would not be weakened by the omission of all mention of docking. Why include the inconclusive docking material when the authors have valid and conclusive experimental data?

With respect to molecular dynamics. I believe that most folks think that relevant conformational changes in allosteric proteins take place on a time scale of milliseconds. The authors have not

explained how the microsecond simulations that they report can be relevant. Isn't the best available set of molecular dynamics potential functions the set developed by D. E. Shaw? It seems that these were not used. Reference 32, from 2013 is used to support the use of molecular dynamics in analyzing an allosteric system. It seems to me that if molecular dynamics were a valuable tool in the analysis of allosteric proteins that there would be a profusion of more recent and definitive studies to cite. The basis of the molecular dynamics analysis done here was the distance between two atoms. The protein was considered to be in one state if the distance were less than 10 Angstroms, and in the other state when the distance was greater than 10. It is pretty hard to see how such a classification method can be closely related to a relevant conformational change in LysG. As in the case of docking, I hope the authors will remove this component of their paper in favor of their solid and convincing experimental data.

Robert Schleif
Professor, Biology and Biophysics
Johns Hopkins University

Reviewer #4 (Remarks to the Author):

In this manuscript, authors solved the structure of LysG protein of *Corynebacterium glutamicum*. The LysG is partially saturated with 12 residues, depending on its structure, to remove its lysine response but to retain its histidine response. And finally explained why the A219L mutation caused lysG to be insensitive to lysine. However, this manuscript is not outstanding in the importance and innovation of this field. There are several concerns to be clarified and I'd like to give several comments which should be addressed.

1. This article is not outstanding in the importance and innovation of structure, because many structures of transcription regulatory factors of LTR family have resolved and they share high structure similarity, and LysG even shares 40% sequence identity with one of them. The structure of LysG and L-arginine complex was determined for the first time, but it is not the key point in this paper. I think if you want to present a detailed structural characterization of the LysG-L-lysine complex and LysG-L-histidine complex, you should determine the structures of them.

As far as I know, LysG regulates gene transcription probably in the form of a tetramer, and it is not clear whether or how small molecules play a role in this process. It's only one-sided to employ molecular dynamics simulations with monomer structure to understand the underlying structure-function relationship.

2. Line 130, when docking proteins and amino acids, five docking tools were mentioned, but why in supplementary Fig. 3 only shows four docking results?

3. Why do you make a percentage of weighted densities? Does this have anything to do with the combination of different ligands? Why don't you just do a correlation analysis of the binding energy

4. In "structure-guided engineering of LysG towards a focused ligand spectrum", 12 amino acids were selected for saturation mutation in LysG to remove the binding capacity of L-lysine based on binding pattern of LysG to L-arginine. LysG does not dock well with L-histidine and L-lysine, so what supports the choice of 12 amino acids? Since the binding sites of L-histidine and LysG are not necessarily the same as those of L-arginine.

5. In fig. 4d, the ratio and distribution of the open state of proteins after the addition of arginine and

histidine are similar, and even the open state of proteins after the addition of arginine is more than that after the addition of histidine, why do proteins bind to histidine and arginine three orders of magnitude differently and respond better to histidine than to arginine

6. The calculation criteria for histidine and lysine in fig. 4d are different from that for arginine. Why compare the results calculated from different criteria?

7. LysG protein responds to histidine, lysine, and arginine. In this study, only sensors that responded specifically to histidine were screened. There are relatively few explanations for arginine. Have you screened arginine Acid specific response biosensor and done some analysis?

Point-by-point response to Reviewers

Manuscript ID: NCOMMS-20-10185-T

Engineering and application of a biosensor with focused ligand specificity

Reviewer #1:

*The authors sought a variant of LysG having reduced sensitivity toward lysine, in hopes of developing a biosensor that would enable high-throughput screening for mutants of *C. glutamicum* producing His. To do this, a crystal structure of LysG was first determined. Various combinations of residues potentially important to ligand binding were randomized and the libraries were screened by FACS to isolate L-lysine-insensitive variants that still respond to L-histidine. “pSenHis” was ultimately isolated and studied by ITC and MD. Finally, the sensor was used to isolate a variety of hisG mutant *C. glutamicum* strains that overproduced His.*

Overall this is a well-written and thorough study that addresses the challenge of TF-based biosensor specificity. The experiments and described in sufficient detail, and statistical analysis is sufficient. My only concerns / questions relate to the ITC and MD results

1. We know that LysG responds well to lysine (essentially the same response as to his at low concentrations) – so why is the measured K_d so different? How can this be rationalized?

Thank you for this question. All ITC-experiments were performed *in vitro* with isolated and purified protein, whereas the (biomass-)specific biosensor response *in vivo* is not only the result of the interaction of LysG with the respective amino acid: LysG is part of the biosensor construct within a living cell. This cell also responds to elevated amino acid (or dipeptide) concentrations supplemented to the medium. Such an effect can be observed in Figure 2 in the case of L-arginine. Elevated intracellular L-arginine levels have a negative impact on the cell metabolism resulting in a decrease of specific fluorescence between 1 and 3 mM dipeptide added. Hence, we are careful when we draw any conclusions from both sets of experiments.

However, we think that the ITC data supports the observed L-lysine insensitivity and the unaltered L-histidine response of the LysG-A219L biosensor variant.

2. Then, the MD results suggest that lysine acts to inhibit gene activation. Shouldn't that binding also be reflected in the ITC data? Why or why not?

Well, LysG always acts as transcriptional activator in *Corynebacterium glutamicum*, inducing gene expression upon binding to the operator region in response to the presence of any of the three basic amino acids acting as inducers. We think that ITC data and MD data support this notion as all amino acids bind to the wild-type regulator LysG during ITC analysis. In the MD experiments, the “open conformation” of the transcriptional regulator is considered to be the transcription-activating conformation, which is stabilized upon interaction with L-lysine.

3. Finally, and related, results for the response of pSenHis in the presence of both his and lysine should be presented. Or some experimental analysis of. Is there clear competitive inhibition?

Thank you for these two suggestions. We indeed performed the first suggested set of experiments (“pSenHis in the presence of both his and lysine”) of which the results can be found in the text and Figure 3. We thought about the second suggested type of experiment (“LysG-A219L in the presence of both ligands”), but since LysG-A219L does not bind L-lysine (as observed during ITC experiments – and indirectly also observed during biosensor characterizations and micro-scale cultivations) and since docking experiments suggest the same binding site for all three ligands, one would not expect any effect in the presence of both ligands.

Reviewer #2:

This paper has shown a semi-rational engineering of a transcriptional regulator protein, LysG, by high-throughput screening of the site-saturated mutants based on the LysG and LysG-Arg high resolution structures. It demonstrates that the mutant A129L could significantly reduce the LysG affinity to L-lysine, which is novel and helps to screen L-histidine-producing strains efficiently.

The findings in this paper are useful for other researchers to continue the rational engineering of LysG for different needs in the amino acid biosensor. It encourages other researchers to spend time to obtain the high-resolution structures to assist the high-throughput screening of the target.

1. The authors obtained LysG-Arg structure but mainly showed the validation of pSenHis rather than pSenArg, which decreases the significance of detailed analysis and tight correlation between the structure and function. To prove that the binding site of His could be same with Arg in LysG, several different data are used to indirectly demonstrate it. However, it's still possible that the mutation site A129L influences other adjacent structures related to His, which behaves differently or even binds in a different site compared with Arg in LysG. And the result, in this paper, that none of C. glutamicum variants can accumulate L-Arg could be one evidence. It means that the mutations based on the analysis of LysG-Arg structure did not help to screen out cell variants for L-Arg accumulation.

Thank you. Of course, we cannot exclude that the amino acid substitution A219L affects other structures within the LysG protein and that another potential binding site exists. However, all docking experiments (consensus running of AutoDock, rDock, LeDock, and DOCK6) as well as the MD-simulations all hint towards a single binding site for all basic amino acids.

Noteworthy in this context, neither with pSenLys (the original biosensor), nor with pSenHis (the engineered variant of this study) could any L-arginine accumulating C. glutamicum variants be isolated from randomly mutated libraries (pSenHis: this study; pSenLys: Binder, S. et al. A high-throughput approach to identify genomic variants of bacterial metabolite producers at the single-cell level. *Genome Biol.* **13**, R40 (2012)).

Important to keep in mind: the original biosensor pSenLys with the wild-type LysG protein was successfully used to isolate L-arginine accumulating variants from libraries in which genes for key enzymes of L-arginine synthesis were specifically(!) mutated – but not from randomly mutated libraries! (Schendzielorz, G. et al. Taking Control over Control: Use

of Product Sensing in Single Cells to Remove Flux Control at Key Enzymes in Biosynthesis Pathways. ACS Synth. Biol. 3, 21–29 (2014)). This information is already included in the introduction (line 83).

Hence, we assume that the metabolic network, which is under stringent metabolic control, is the reason for not obtaining any L-arginine producing variants. The likelihood to obtain L-lysine producing variants proved to be easiest - This is no surprise as this pathway is quite isolated in the metabolism. In contrast, the L-arginine pathway is intertwined with (and dependent on) L-glutamate-, L-ornithine- and L-aspartate biosynthesis and their individual genetic regulation. Furthermore, there is also a direct link to the nitrogen metabolism. Hence, the likelihood to find an L-arginine producing variant is much lower.

Last but not least, we would like to stress that we never set out to engineer an L-arginine specific biosensor and that we never intended to isolate L-arginine producing variants.

2. Regarding to the positioning of Arg in the electron density map, it's not persuasive with the open-closed conformation analysis by MD. If the researcher can explain from other aspects about why Arg is not positioned 180° rotated, it will be better. D124 and E125 look to be more competitive than D193 to interact with the positively charged guanidino group of Arg. The whole demonstration could be better if the authors could address above questions.

This is a very good question! At the limited resolution of the electron density map, this distinction is not straightforward. Of course, we did place the effector only after careful refinement of the LysG structure alone. While the resulting bias-free difference density clearly favored the orientation shown in Fig S1, we tried to refine both orientations, but came up with the same preference. The simulated annealing omit map of the ligand using the fully refined model again supports this orientation. As reported in L292 ff, we did investigate this further using independent simulations with L-arginine placed both in the reported conformation and 180 degrees rotated. In the rotated case, L-arginine reoriented back to the reported conformation within the first few nanoseconds. This is unlikely to be a spurious effect. In combination, we think that this provides convincing evidence for the choice in orientation.

Comments/questions of Reviewer #2 in the manuscript:

3. line 131: it is better to not include A219L mutants if it's used here

Thank you for the suggestion. We agree that the reader is not aware of the A219L mutation at this point, but given the positioning of this figure in the Supplementary Information, we consider it sufficient to add additional descriptive text in the legend that indicates where A219L is described in the main text. Therefore, we added “*for the wildtype LysG, as well as for the engineered LysG_A219L (see The pSenHis biosensor is L-lysine-insensitive).*” to the legend of supplementary figure 3.

4. line 153: it will be better to discuss here or later why pairwise mutagenesis is used. More efficient?

Thank you for this question. Multi-site directed mutagenesis is done to dramatically increase the number of protein variants to screen (in this case: 12 positions x 32 codons (NNS) = 768 variants (single site saturation mutagenesis) vs. 6 PCRs x (32 codons x 32 codons = 1024) = 6144 variants (multi-site directed mutagenesis). This is usually done, when the experimentalist is not limited by the throughput of the employed screen. This is the case here since we used FACS. Noteworthy, only when screening such libraries with more

than one codon targeted, synergistic mutations can be discovered. This is an additional advantage of pairwise mutagenesis.

For more clarity, we included an explanation, directly in the results section:
“These twelve residues were mutated pairwise (N95+R143; D97+H161; D124+E125; F189+D193; G190+P191 and A219+F222) by multi-site directed saturation mutagenesis to increase the protein sequence space and to directly take advantage of potentially synergistic effects of neighboring amino acid substitutions.”

5. line 187: is it C565T?

Thank you for noticing. Its C656T, the error was introduced one sentence earlier. We corrected the sentence to: *“Whereas the T666C-transition was silent (F222F), the C656T transition did cause an A → L amino acid substitution at position 219.”*

6. line 283: it could be better to understand if this analysis is discussed with the purpose to reduce the L-lys binding affinity.

Thank you for this suggestion. Upon rereading and discussing alternative formulations among the authors, we decided to leave the analysis as is. It sounds more natural as we discovered this useful point mutation and did not design it. However, we close the circle with the last two sentences of this paragraph (line 298-300).

7. line 304: both samples have

True indeed. We corrected the sentence to: *“Both samples have an open (blue) and closed (tan) conformation of the ligand binding domain, where openness is defined by distance $>10\text{\AA}$ between atoms 96 C α and 219 C α .”*

Reviewer #3 (Robert Schleif):

I support the publication of this paper--with some modifications. It describes the semi-rational engineering of a desired specificity change of a protein and the utilization of this changed specificity to isolate a mutant bacteria overproducing the ligand of the engineered protein. The approach and findings should be of interest to a wide audience. My comments below are of two types. The first concern relatively minor changes that improve readability, understanding, or accuracy, and the second concern removing the docking and molecular dynamics components of the paper which I find to be unjustified and unsatisfactory.

First type of comment:

1. l. 74 *In my experience, the vast majority of transcriptional regulators can be characterized as having a narrow specificity, not broad. The authors' characterization of known transcriptional regulators simply could to be changed to state that in a good number of cases, the ligand specificity of a transcriptional regulatory protein needs to be altered in order to permit its efficient utilization in a sensor system.*

Thank you, we agree! We changed the sentence to: *“A major challenge which has not been tackled yet is the relaxed ligand specificity of some TRs, limiting their efficient utilization in a biosensor system.”*

2. l. 87 *The intended meaning is more clearly communicated with the omission of "large".*

Corrected: *“The reason for this observation is still unclear, possibly the dissociation constant (KD) for L-histidine and L-arginine is too high compared to the intracellular abundance of these amino acids, making changes in intracellular concentrations of these amino acids difficult to detect.”*

3. 197-100 *It would really help the reader to state that the transcriptional regulator stimulates activity of a promoter that drives the synthesis of a fluorescent protein, which allows fluorescence activated cell sorting of individual cells. Also, at its first appearance in the paper, the abbreviation eyfp needs to be spelled out (l 138?).*

Thank you. We included the suggested explanation along with the introduction of the abbreviation eyfp at the end of the introduction: *“The engineered LysG variant with the narrowed ligand spectrum was subsequently used to construct the biosensor pSenHis. As part of this sensor, the transcriptional regulator stimulates activity of a promoter that drives the synthesis of the fluorescent protein EYFP (Enhanced Yellow Fluorescent Protein), which in turn allows fluorescence-activated cell sorting of individual cells. This new biosensor was studied on the single-cell level using microfluidics and successfully applied in a FACS-based screening of 10⁷ chemically mutagenized C. glutamicum wild type cells for identifying L-histidine producing C. glutamicum variants.”*

4. 1. 109-110 *It sounds contradictory to state that the crystal structure of LysG had to be solved because there were no suitable homologs, and then to use molecular replacement to solve the structure. Perhaps the authors could tell us retrospectively whether it really was necessary to solve the crystal structure.*

Thank you for this excellent question. A homologue suitable for molecular replacement might only share 20% sequence identity if the fold is conserved, but for full atomistic modeling of side chain positions, substantially higher sequence identity would be necessary. This is especially true for predictions regarding effector/ligand binding sites. For LysG, no homologue could be found that would have yielded a sufficiently precise model (the closest related structure (ArgP) shares around 40% identical residues). This study aims at the discussion of detailed ligand interactions, for which a crystal structure is a necessary starting model in absence of an excellent homology model. In addition, the structure with bound effector strengthens the discussion by delivering experimental evidence.

In response to this particular question of the reviewer, we calculated a LysG homology model based on the ArgP-structure. While the overall fold is well represented of course, the modelled binding site geometry suffers from backbone shifts of up to 1-2Å and varying sidechain orientations. This would not have been a good starting point for the selection of residues to be targeted by multi-site directed mutagenesis or any simulation.

5. 1. 260 *Please provide the detection limit.*

Thank you! Information included.

6. 1. 359 *Wasn't the redesign of AraC also structure-based?*

Yes, this is true for the studies from the Cirino lab concentrating on AraC reengineering. The LacI engineering project also mentioned in this sentence was mostly done in a random manner using protein-wide single-amino-acid saturation mutagenesis and error-prone PCR. **We rephrased the following sentence:** *“However, compared to the LacI studies where the TR was mutagenized by protein-wide single-amino-acid saturation*

mutagenesis and error-prone PCR, LysG was carefully redesigned by structure-guided and pairwise mutagenesis of selected residues in the ligand binding site prior to FACS.”

7. I.386 *But plenty of transcription factors have been studied in far more detail and depth that was done here for LysG. The depth of study and the coupling to a promoter and fluorescent protein are two independent operations.*

Yes we agree, and rephrased the sentence: *“The steric extent of L-histidine and the more restricted flexibility compared to L-lysine explain why a single amino acid substitution can decrease the ligand spectrum of this TR, allowing for the construction of a biosensor with a focused ligand spectrum.”*

8. *In this same paragraph, one can't help wondering why the structure of LysG wasn't predicted using David Baker's approach of coupling the sequences of a large number of homologs to a prediction by Rosetta.*

Even though computational protein structure prediction and homology modeling have recently become more reliable, at the time when this experiment was performed, it was not conceivable to obtain a structure of LysG with sufficient detail from homology modeling. In absence of sufficiently high sequence identity of solved structures and with the aim to obtain a high-resolution binding site conformation, the experimental structure determination was absolutely necessary (see also the answer to comment 4 of Reviewer #3).

9. *I believe it would enhance the paper to describe more carefully and completely how it is first of all, that LysG can respond to lysine, arginine, and histidine, and then why the mutant can no longer respond to lysine while still responding to the other two amino acids. The authors made a brief stab at part of this, but it was too hard for me to follow.*

Thank you for this suggestion. We have carefully reconsidered our argument based on the MD analysis and reformulated the following part of the text to be more precise: *“The steric extent of L-histidine and the more restricted flexibility compared to L-lysine explains why a single amino acid substitution can decrease the ligand spectrum of this TR, allowing for the construction of a biosensor with a focused ligand spectrum”.*

10. *And now, the arguments for removal of all mention of docking and molecular dynamics. With respect to docking. First note the authors' statement at the end of the legend of Fig. S 3. “We find that the docking tools do not agree very well amongst each other, and none of the findings provide a conclusive argument for alternative binding poses for L-histidine.” Such a statement invalidates the statement made elsewhere in the paper relating to the number of water molecules released from the protein upon one ligand's binding. The precise use of the docking programs is not described and it does not appear that others in the field could exactly reproduce what the authors claim. Also, if such docking programs are effective, I would expect them to be used to find and improve potential drugs against medically important proteins. Are they?*

This is an interesting remark. The current literature has many prominent examples of successful application of docking programs in in-silico drug discovery procedures. Indeed, within Pharma industry, a strong shift towards docking based library screening can be observed. Below are some insightful review papers from the last 4 years that provide ample evidence for the scientific validity of such tools. It is also important to highlight that our study combines docking with MD simulations and that the ligands have not diffused substantially

from the proposed binding site during the relatively long simulation times. This indicates that Docking Scoring Functions, as well as Classical Protein Force Fields align in this study.

Ahsan, M. J., Choupra, A., Sharma, R. K., Jadav, S. S., Padmaja, P., Hassan, M., ... & Bakht, M. A. (2018). Rationale design, synthesis, cytotoxicity evaluation, and molecular docking studies of 1, 3, 4-oxadiazole analogues. *Anti-Cancer Agents in Medicinal Chemistry (Formerly Current Medicinal Chemistry-Anti-Cancer Agents)*, 18(1), 121-138.

Pagadala, N. S., Syed, K., & Tuszynski, J. (2017). Software for molecular docking: a review. *Biophysical reviews*, 9(2), 91-102.

De Vivo, M.; Masetti, M.; Bottegoni, G.; Cavalli, A. Role of Molecular Dynamics and Related Methods in Drug Discovery. *J. Med. Chem.* 2016, 59, 4035–4061.

11. Finally, the paper would not be weakened by the omission of all mention of docking. Why include the inconclusive docking material when the authors have valid and conclusive experimental data?

Thank you for this suggestion. We discussed this but came to the conclusion that our current claims are in alignment with the level of credibility that can be attributed to docking programs. In order to start an MD simulation, it is necessary to define a starting configuration of the protein+ligand system. In absence of crystal structures for LysG+Lys and LysG+His, it is necessary to place the ligands in a reasonable starting position. Even though docking tools might not place a ligand exactly in the correct position, the rmsd variance between the best scoring docking poses of the used docking tools was very small. In addition, the MD simulations did not diffuse the ligands away from the proposed docking site, providing additional evidence for the correct identification of the binding site. Given that Docking was solely used to identify the position of the substrates in the binding site, and that it is a necessary part of our MD method, we did not remove this content.

11. With respect to molecular dynamics. I believe that most folks think that relevant conformational changes in allosteric proteins take place on a time scale of milliseconds.

Thank you for this remark. Given that the simulations start with the ligand in the active site of the protein, we do not need to simulate the diffusion of the ligand into the active site. Instead, this study focusses on the conformational changes induced by the binding event, which occur on the nanosecond timescale. The best evidence for this is that these transitions were frequently observed during the various simulations as reported here. This is quite common for small conformational changes. Larger conformational changes, like ion channel openings or folding of globular proteins occur on the millisecond timescale instead. Our findings of relevant protein motion on the nanosecond timescale are in agreement with our expectation and current consent in the field. This article (in particular figure 1 therein):

Dror, Ron O., Jensen, Morten Ø., Borhani, David W., Shaw David E. (2010). Exploring atomic resolution physiology on a femtosecond to millisecond timescale using molecular dynamics simulations. *J Gen Physiol.* 2010 Jun; 135(6): 555–562
(<https://www.ncbi.nlm.nih.gov/pmc/articles/PMC2888062/>)

12. The authors have not explained how the microsecond simulations that they report can be relevant.

This is a good point. In the original submission, we omitted a specific argument. But given the broad audience of this journal, **we added the following explanation and citation:**

“To gain additional insights into ligand-receptor interaction, extensive MD in the microsecond time range was performed for LysG and LysG-A219L with and without L-histidine, L-lysine and L-arginine in the ligand binding site to reveal conformational changes that typically occur on the nanosecond timescale [REF].”

[REF: Wapeesittipan, P., Mey, A. S. J. S., Walkinshaw, M. D. & Michel, J. Allosteric effects in cyclophilin mutants may be explained by changes in nano-microsecond time scale motions. Commun. Chem. 2, 1–9 (2019).]

13. *Isn't the best available set of molecular dynamics potential functions the set developed by D. E. Shaw? It seems that these were not used.*

Indeed, the Amber99sb-ildn forcefield with torsion optimizations by D.E.Shaw was used in this study. In order to avoid confusion, we added the following citation to the online methods (computational modelling): Lindorff-Larsen, K. et al. Improved side-chain torsion potentials for the Amber ff99SB protein force field. Proteins Struct. Funct. Bioinforma. 78, 1950–1958 (2010).

14. *Reference 32, from 2013 is used to support the use of molecular dynamics in analyzing an allosteric system. It seems to me that if molecular dynamics were a valuable tool in the analysis of allosteric proteins that there would be a profusion of more recent and definitive studies to cite.*

Thank you for this remark. Indeed, reference 32 focusses on a very specific allosteric event. We expanded the citation list by two excellent reviews and a more recent study to represent the large corpus of MD based papers studying allosteric transitions in proteins:

Wapeesittipan, P., Mey, A. S. J. S., Walkinshaw, M. D. & Michel, J. Allosteric effects in cyclophilin mutants may be explained by changes in nano-microsecond time scale motions. Commun. Chem. 2, 1–9 (2019).

Hertig, S., Latorraca, N. R. & Dror, R. O. Revealing Atomic-Level Mechanisms of Protein Allostery with Molecular Dynamics Simulations. PLoS Computational Biology 12, (2016).

De Vivo, M., Masetti, M., Bottegoni, G. & Cavalli, A. Role of Molecular Dynamics and Related Methods in Drug Discovery. Journal of Medicinal Chemistry 59, 4035–4061 (2016).

15. *The basis of the molecular dynamics analysis done here was the distance between two atoms. The protein was considered to be in one state if the distance were less than 10 Angstroms, and in the other state when the distance was greater than 10. It is pretty hard to see how such a classification method can be closely related to a relevant conformational change in LysG.*

Thank you for this comment. The relevant reaction coordinate of the proposed mechanism is the opening of the two domains. The determined distance describes exactly this reaction coordinate. **As this was not explicitly stated in the manuscript, we made the following addition:** *“Two 1 μ s simulations of LysG revealed two frequently occupied conformations with a reaction coordinate that can be exactly described by the distance between C α atoms of residues 219 and 96, which are 6.4 Å apart in the effector-occupied (closed) RD in the LysG-Arg structure (Fig. 4a).”*

16. *As in the case of docking, I hope the authors will remove this component of their paper in favor of their solid and convincing experimental data.*

We appreciate the thoughtful comments and suggestions throughout. We believe that the additional clarifications and explanations will make this manuscript more

accessible to a broader audience. Given that the atomistic behavior observed in the MD provides the best accessible model to explain the experimental data, we kindly disagree with this particular suggestion and prefer to retain the MD data in the text.

Reviewer #4

In this manuscript, authors solved the structure of LysG protein of Corynebacterium glutamicum. The LysG is partially saturated with 12 residues, depending on its structure, to remove its lysine response but to retain its histidine response. And finally explained why the A219L mutation caused lysG to be insensitive to lysine. However, this manuscript is not outstanding in the importance and innovation of this field. There are several concerns to be clarified and I'd like to give several comments which should be addressed.

1. This article is not outstanding in the importance and innovation of structure, because many structures of transcription regulatory factors of LTTR family have resolved and they share high structure similarity, and LysG even shares 40% sequence identity with one of them. The structure of LysG and L-arginine complex was determined for the first time, but it is not the key point in this paper. I think if you want to present a detailed structural characterization of the LysG-L-lysine complex and LysG-L-histidine complex, you should determine the structures of them.

Thank you for this statement. It is often an experimental challenge to obtain crystal structures of highly flexible proteins with ligands. In particular, transcriptional regulators, which undergo large conformational changes such as LysG are challenging. Our efforts to crystallize LysG in complex with Lys and His were unfortunately unsuccessful and therefore not reported here.

2. As far as I know, LysG regulates gene transcription probably in the form of a tetramer, and it is not clear whether or how small molecules play a role in this process. It's only one-sided to employ molecular dynamics simulations with monomer structure to understand the underlying structure-function relationship.

Thank you for this comment. Indeed, LysG regulates in the form of a tetramer. However, the regulatory domains containing the binding site trigger the conformational changes in the tetramer. Therefore, we assume the four binding domains in the tetramer to exhibit independent allosteric motion. It is commonly accepted that small changes in the protein conformation can lead to substantially different regulatory behavior and that small molecules are able to induce the necessary allosteric motions.

3. Line 130, when docking proteins and amino acids, five docking tools were mentioned, but why in supplementary Fig. 3 only shows four docking results?

Thank you for noticing. We employed SwissDock in the early stages of this project to identify possible mutation sites, as shown in Fig. 5. The other four docking tools were used to perform a consensus docking to position the substrates for MD simulation. We removed SwissDock reference therefore from line 130 (now l.132), but retained it in line 467 (now l. 464).

4.. Why do you make a percentage of weighted desities? Does this have anything to do with the combination of different ligands? Why don't you just do a correlation analysis of the binding energy

Thank you for these questions. Consensus docking tries to reduce the inherent error in a single docking tool by comparing the outcomes of multiple tools amongst each other. In all cases only a single ligand is docked on a protein, there are no combination effects investigated. The results of a docking simulation are docking poses, which are sorted by binding energy. Docking tools will predict binding sites with low energies all over the protein, but the correct binding site will be populated with high energy poses. The aim of this simulation is therefore to identify, which fraction of docking poses with high binding energies falls into the Arginine binding site from the crystal structure. The easiest metric to compare the relative occupation of the L-arginine binding site by proposed docking poses is therefore a calculation of the weighted densities as described. A simple correlation analysis of the binding energy would not contain the same amount of information.

5. In "structure-guided engineering of LysG towards a focused ligand spectrum", 12 amino acids were selected for saturation mutation in LysG to remove the binding capacity of L-lysine based on binding pattern of LysG to L-arginine. LysG does not dock well with L-histidine and L-lysine, so what supports the choice of 12 amino acids? Since the binding sites of L-histidine and LysG are not necessarily the same as those of L-arginine.

Thank you for this comment. The twelve residues in the ligand binding site were selected based on the structure model. In addition to eight residues (N95, D97, D124, E125, H161, F189, D193, and F222) with direct contact to the amino acid ligands, four second shell residues were included, which are not strictly conserved in LTTR-type TRs (R143, A219, G190, P191). This information can be found in the first paragraph of the subchapter "Structure-guided engineering of LysG towards a focused ligand spectrum" (starting at line 148).

There must have been a slight misunderstanding with regards to the consensus docking results as reported in L568. In fact, all docking tools suggest similar binding sites for the three basic amino acids. The MD simulations also provide evidence that the LysG Ligand complexes are stable for all three ligands placed in the L-arginine binding site. Of course, we cannot completely rule out that other residues might also play a role, but we regard our choice of target residues as well substantiated.

6. In fig. 4d, the ratio and distribution of the open state of proteins after the addition of arginine and histidine are similar, and even the open state of proteins after the addition of arginine is more than that after the addition of histidine, why do proteins bind to histidine and arginine three orders of magnitude differently and respond better to histidine than to arginine

This is a very good question. Indeed, ITC suggests that L-histidine binds 1,000 times stronger than L-arginine. Also, the fluorescence images show that L-histidine results in a stronger signal than L-arginine. Comparing the histograms, it is evident that the distribution is strongly shifted for LysG+ L-lysine compared to LysG_A219L+ L-lysine. The difference between His and L-arginine is not as evident from the histogram representation. However, as described in the text, the LysG+ L-arginine complex is very stable in the open and closed conformations, dependent on which starting conformation was selected. Because we did not observe transition on the nanosecond timescale from either conformation it is argued that the energy barrier for this complex is rather high. This provides a solution to the above described paradox. The histograms look similar, because we sample from two different conformations, but no transitions are observed. Therefore, the dynamic of the LysG+ L-arginine complex is much slower and results in less signal once formed. In order to verify the binding energy

difference, free energy calculations would be necessary, but these are beyond the scope of this manuscript.

This already included in the main text (295 ff):

“For the perturbed starting configuration also, no transition into the stable closed conformation was observed, suggesting that the energy barrier between both states is higher than for the TRs by themselves or in complex with L-histidine or L-lysine. This corresponds well with the successful crystallization of only the LysG-L-arginine complex and verifies the correct positioning of L-arginine. Further, the high energy barrier explains why fluorescence for both TRs in presence of L-arginine was reduced compared to the other ligands.”

7. *The calculation criteria for histidine and lysine in fig. 4d are different from that for arginine. Why compare the results calculated from different criteria?*

Thank you for this question. It is not exactly clear what difference this refers to. In all cases, we used the same criteria: MD simulation, measurement of distance between residues 219 and 96, calculation of fraction below and above 10 Å. For L-arginine we started simulations from the closed and open conformation, as no transition was observed. Similar, we started LysG_A219 + L-lysine complex from the closed and open conformation. For LysG_A219 + L-lysine complex simulation we observed a rapid transition from open back into the closed conformation. Therefore, we concluded that LysG_A219 + L-lysine does not stabilize the open conformation, while L-arginine stabilizes open and closed conformations of LysG and LysG_A219.

8. *LysG protein responds to histidine, lysine, and arginine. In this study, only sensors that responded specifically to histidine were screened. There are relatively few explanations for arginine. Have you screened arginine Acid specific response biosensor and done some analysis?*

Thank you for this question! Our motivation to develop an L-histidine biosensor was driven by the desire to provide a tool for rapidly engineering microorganisms towards L-histidine production. Hence, we did not screen for an L-arginine-specific biosensor as part of this project. However, with knowledge and the L-lysine-insensitive biosensor (pSenHis) we have now in our hands, it would be very interesting to follow two possible routes towards an L-arginine biosensor: (1) Start from pSenHis and try to focus the ligand specificity solely on L-arginin, or (2) Start from the original biosensor pSenLys to see if another evolutionary route (avoiding position 219) is necessary to reach this goal. However, considering the time and man-power needed for this project, this is something for the future.

Review Comments

Reviewer #2 (Remarks to the Author):

The authors have answered the main questions raised in my first review.

Considering the purpose of the authors to engineer an L-Histidine specific biosensor with the available LysG-Arg structure, I agree that this work is successful in its strategy on isolating l-histidine-producing strains by FACS in high throughput. The possibility that the L-Arg production is more complicated and different from L-Lys and L-His could be accepted to explain the difficulty to find an L-Arg producing variant based on LysG-Arg structure.

The positioning of Arg is well-explained by the authors with their previous attempts to model different rotations, although it still looks a little weird.

The authors have revised the errors in the main manuscript and provide good explanation to the questions.

Reviewer #3 (Remarks to the Author):

This is a review of the revised manuscript.

Summary: I remain unconvinced that retention of the molecular docking or molecular dynamics portions of the paper are important to the paper, have been adequately performed, or that publication of this portion of the material meets the standard publication standard of being reproducible by other experts in the field as both the docking and MD have been incompletely described.

Without accounting for flexibility in the protein and the ligands, I do not see how the docking results say anything more than that the ligand was put in a cavity in the protein. If the results meant something more, wouldn't the ligands have been positioned by the various programs to within 0.2 Angstroms of one another? Additionally, computational trials with ~25 similar sized ligands would have confirmed specificity of ligand binding.

I feel that molecular dynamics runs of 2 to 250 nanoseconds as were used here seem unlikely to reveal important allosteric conformational changes. The authors added several references in support of their very short simulation times. In fact, the references do not support the validity of such short simulations. Here are three statements taken from the newly added references. They all indicate that in general very much longer simulations would be required to study allosteric changes.

"The binding of certain ligands, however, takes place on timescales too slow to be captured by simulation, even on supercomputers. In some cases, one can circumvent this problem by using large numbers of short simulations together with statistical modeling techniques that describe long-timescale events as sequences of short-timescale events."

"A straightforward approach to studying the β 2AR activation mechanism by MD would be to initiate simulations from the receptor's inactive state, with an agonist placed in the orthosteric site. Unfortunately, the estimated timescales of receptor activation are many milliseconds, substantially beyond the timescales that were accessible to MD simulations."

"We found, however, that simulations of an agonist-bound receptor initiated from the active state, with the G protein removed, transitioned spontaneously to the inactive state within a few microseconds." Note, a transition in a few microseconds, not not tens of transitions in 0.25 microseconds as is implied in the submitted manuscript.

To some of the original reviewer's comments the authors responded in their letter accompanying the

revised manuscript, but did not address the comments within the manuscript. If the reviewer is confused, others probably will be as well, and it would be better if clarification or acknowledgment were in the paper itself.

Robert Schleif

Reviewer #4 (Remarks to the Author):

I satisfy the answers from the authors.

Point-by-point response to Reviewers

Manuscript ID: NCOMMS-20-10185A

Engineering and application of a biosensor with focused ligand specificity

Reviewer #2:

The authors have answered the main questions raised in my first review.

Considering the purpose of the authors to engineer an L-Histidine specific biosensor with the available LysG-Arg structure, I agree that this work is successful in its strategy on isolating L-histidine-producing strains by FACS in high throughput. The possibility that the L-Arg production is more complicated and different from L-Lys and L-His could be accepted to explain the difficulty to find an L-Arg producing variant based on LysG-Arg structure.

The positioning of Arg is well-explained by the authors with their previous attempts to model different rotations, although it still looks a little weird.

The authors have revised the errors in the main manuscript and provide good explanation to the questions.

-Thank you!

Reviewer #3 (Robert Schleif):

A.

This is a review of the revised manuscript.

Summary: 1) I remain unconvinced that retention of the molecular docking or molecular dynamics portions of the paper are important to the paper, 2) have been adequately performed, 3) or that publication of this portion of the material meets the standard publication standard of being reproducible by other experts in the field as both the docking and MD have been incompletely described.

→ Thank you for the additional remarks. We will respond point by point:

- 1) The experimental findings of this study suggested possible alternative binding sites for ARG, LYS, and HIS. We therefore performed docking calculations to investigate the potential presence of alternative binding sites. The docking results suggest that all substrates dock into the same binding site and not on the surface as we speculated earlier in the project. This is a very useful finding, thus we think is absolutely worth presenting. Further, for the subsequent MD simulations, the substrates have to be placed into the most probable position and molecular docking is the most reproducible way of achieving this goal.

2) With regard to the quality of the application of the methods, we have ensured highest confidence by using multiple docking tools, with flexible ligand docking, on different protein conformations to account for protein dynamics. The coupling of this procedure with MD is state of the art, so we cannot agree with the assessment of the calculations being in-adequately performed.

3) **We agree that parts of the method were insufficiently described and used this opportunity to add further detail to the “Computational Modelling” section in the “Online Methods” section.** Furthermore, we simplified the docking analysis for this publication and will make the full consensus docking method available in a separate publication. **The scripts necessary to reproduce the updated docking procedure are uploaded to a persistent repository (see below).**

The reproducibility of MD results is traditionally ensured by provision of runtime files with structural coordinates, velocities, and all necessary simulation parameters. We have uploaded the runtime files for each of the MD runs to a persistent repository at SimTK (<https://simtk.org/projects/lysg>). SimTK is a free project-hosting platform funded by the NIH and is part of the Simbios project. The open and free GROMACS software (www.gromacs.org) can be used to reproduce the reported trajectories.

B.

1) *Without accounting for flexibility in the protein and the ligands, 2) I do not see how the docking results say anything more than that the ligand was put in a cavity in the protein. 3) If the results meant something more, wouldn't the ligands have been positioned by the various programs to within 0.2 Angstroms of one another? 4) Additionally, computational trials with ~25 similar sized ligands would have confirmed specificity of ligand binding.*

→ **Thank you, we will respond point by point:**

- 1) The flexibility of the protein was in fact accounted for by docking on an ensemble of protein structures obtained by MD simulation, the ligands are flexible in the docking procedure itself (the default of all modern docking tools, like AutoDock). **We have described this in more detail in the “Computational Modelling” section in the “Online Methods” section.**
- 2) This statement captures the main results of all docking calculations. The purpose of a docking calculation is the identification of potential binding sites (which are typically some cavities that fit the ligand that are then ranked according to a scoring/energy function). **To make this clearer, we rephrased the following sentence:** “*Consensus docking calculations suggested that L-histidine and L-lysine bind in the same active site as L-arginine (Supplementary Fig. 3).*” (page 13, line 299) to “*Docking calculations suggested that L-histidine and L-lysine bind in the same binding pocket as L-arginine and provided starting conformations for MD.*”
- 3) **This statement surprises us.** No standard docking tool would commonly place ligands with an accuracy of 0.2 Å. Even after alignment of multiple docking poses, it is common to find 1 Å RMSD between heavy atoms of two poses, simply due to ligand flexibility. In evaluations and comparisons of docking tools, oftentimes 2 Å RMSD is considered a successful binding prediction. Our line of argument was that the docking placed the ligands repeatedly into the same binding pocket, independent of protein

conformation as sampled from MD, from which we conclude that surface binding is unlikely.

- 4) Specificity of ligand binding is difficult to determine and requires binding energy calculations (which are not accurate). This is a different question from the one we set out to answer. Our aim was the identification of a binding site for ligands that bind according to experimental results. For the investigated ligands we know that binding does happen, so the problem simplifies to the question of finding the binding site. If surface binding ligands would be known for this protein the suggested procedure would be a useful test.

While our original docking procedure is useful and provides additional insights, we agree that the current description is not sufficient to do the method justice. We therefore decided to publish a separate method paper on the consensus docking method. For the present paper **we reduced the complexity of the docking and removed the analysis of docking results on an ensemble of protein structures with multiple tools from the manuscript and the SI**. This does not change any of the results presented in this paper, as the binding site is sufficiently validated by crystal structure docking with AutoDock. Further, the starting points for MD were taken from docking on the crystal structure alone. **Therefore, this change only simplifies the paper, without changing any of the results or conclusions.**

As a result, we have added details on the docking procedure to the Methods section, and we removed supplementary figure SI3 along with the legend. In addition several sentences in the main text were changed and three references referring to the tools DOCK6, LeDock, and rDock were removed:

- page 6, line 130ff: *“However, since no diffracting LysG-crystals in the presence of L-lysine or L-histidine could be obtained, blind docking calculations using the [LysG+Arg] structure and AutoDock Vina²¹ were performed, which predicted the positions of L-arginine, L-lysine, and L-histidine to be in the same binding pocket with a maximum center of mass deviation of the top 10 docking poses from the crystal L-arginine of 1.2 Å, 1.2 Å, and 1.3 Å, respectively.”*

- page 11, line 250ff: *“While endothermic interactions are frequently coupled to the release of solvent molecules from the protein surface²⁵, blind docking analysis as well as extended MD simulations could not support this hypothesis.”*

C.

1) *I feel that molecular dynamics runs of 2 to 250 nanoseconds as were used here seem unlikely to reveal important allosteric conformational changes. 2)* *The authors added several references in support of their very short simulation times. In fact, the references do not support the validity of such short simulations. Here are three statements taken from the newly added references. They all indicate that in general very much longer simulations would be required to study allosteric changes.*

“The binding of certain ligands, however, takes place on timescales too slow to be captured by simulation, even on supercomputers. In some cases, one can circumvent this problem by using large numbers of short simulations together with statistical modeling techniques that describe long-timescale events as sequences of short-timescale events.”

"A straightforward approach to studying the β 2AR activation mechanism by MD would be to initiate simulations from the receptor's inactive state, with an agonist placed in the orthosteric site. Unfortunately, the estimated timescales of receptor activation are many milliseconds, substantially beyond the timescales that were accessible to MD simulations."

"We found, however, that simulations of an agonist-bound receptor initiated from the active state, with the G protein removed, transitioned spontaneously to the inactive state within a few microseconds." Note, a transition in a few microseconds, not tens of transitions in 0.25 microseconds as is implied in the submitted manuscript.

→ We will respond point by point:

- 1) We do see transitions from open to closed conformations in our simulations, this fact cannot be refuted. The motion thus happens on the timescale of our simulations. Given that the protocol uses standard MD force fields, full atom representations, explicit solvent, and physiological salt concentrations, we think the data should be trusted more than just intuition.

Given that the hinge motion is centered around the binding site, it is not surprising that ligand binding affects the natural motion of the protein on the ns-timescale. This is evident from the fact that simulations for the ligand-protein complex were run for 250 ns in the cases of LYS and HIS and multiple transitions were still observable. For ARG, 2 times 250 ns were required, because it needed to be started from an open and closed conformation, due to the higher energy barrier between the two states.

These distributions show clearly that ligand binding perturbs the unbound energy landscape sufficiently to cause a shift in the sampled conformations. The result that LysG-A219L only samples a closed conformation and instantly returns to it when started from an open conformation is an important finding. It is the only atomistic insight into the regulator function and dependency on ligand binding currently available. As such, it does extend the experimental findings and should be considered an important result to report.

The fact that the reviewer expects much longer timescales could be due to a misunderstanding: The motion we study is not an allosteric motion itself. We simply look at a small two-domain protein with a ligand binding site directly located at the hinge between the two domains. The ligand binding site is referred to as the "allosteric binding site", since the allosteric effect is the activation of the transcription factor (in the DNA-binding domain), which happens further away as a result of the domain motion. But that allosteric effect is not explicitly studied in the simulation. The simulation just contained the regulator domain.

We added the following text to the manuscript, to clarify this point: *"While allosteric transitions frequently occur on longer timescales, the observed transitions are impacted by ligand binding at the hinge region of the two-lobe regulator domain. It should be noted that the simulation does not describe the allosteric conformational change. The simulation only considers the conformational motion of the two-lobe regulator domain and how it is influenced by different ligands. The resulting allosteric effect is the activation of the transcription factor in the DNA-binding domain, which*

happens further away as a consequence of this domain motion, and is not studied here.” (page 17, line 394ff).

- 2) The cited statements all stem from this publication: Hertig *et al.*, *PLoS Comput Biol.* (2016) 12(6): e1004746. (<https://www.ncbi.nlm.nih.gov/pmc/articles/PMC4902200/>). The citations provided are very much in agreement with our computational setup. The first one states:

"The binding of certain ligands, however, takes place on timescales too slow to be captured by simulation, even on supercomputers. In some cases, one can circumvent this problem by using large numbers of short simulations together with statistical modeling techniques that describe long-timescale events as sequences of short-timescale events."

This quote should be read in the context it was written in. The preceding paragraph discusses the time necessary for ligands to diffuse into a protein, given their initial placement in solvent around the protein. The next sentence of this paragraph is more relevant for our discussion: “Alternatively, if the binding site is known, one can position the ligand within the binding site at the beginning of the simulation, allowing its pose and the local protein conformation to adapt during the simulation”. **This is what we have done. By placing the ligand into the binding site, we substantially reduced the necessary simulation time.**

"A straightforward approach to studying the β 2AR activation mechanism by MD would be to initiate simulations from the receptor's inactive state, with an agonist placed in the orthosteric site. Unfortunately, the estimated timescales of receptor activation are many milliseconds, substantially beyond the timescales that were accessible to MD simulations." (These sentences directly follow each other)

"We found, however, that simulations of an agonist-bound receptor initiated from the active state, with the G protein removed, transitioned spontaneously to the inactive state within a few microseconds."

(Note, a transition in a few microseconds, not tens of transitions in 0.25 microseconds as is implied in the submitted manuscript.)

First of all, the cited paper is about a completely different protein (a membrane protein). The protein we consider is a smaller two domain protein. There is no reason to assume that these completely different motions of two very different proteins occur on exactly the same timescale.

But again, we think the **main reason for the reviewer to expect different timescales is the misunderstanding about the allosteric transition as we do not simulate an allosteric conformational change** (as discussed above). Our paper reports on a simulation that indeed did show changes in conformations on the simulated timescale. It is most reasonable to investigate receptor behavior with MD and our protocol is in agreement with best practices. We therefore strongly suggest retaining the MD part in the paper as it does provide additional atomistic detail not contained in the experimental data. **However, as stated above, we expanded the discussion to make sure that readers think in the right direction.**

D.

To some of the original reviewer's comments the authors responded in their letter accompanying the revised manuscript, but did not address the comments within the manuscript. If the reviewer is confused, others probably will be as well, and it would be better if clarification or acknowledgment were in the paper itself.

→ **Thank you.** We extracted all remarks (and respective answers/explanations) of the first revision, which did not result in changes to the main text and added some more information to the main text. (original numbering of the comments kept for orientation).

4. I. 109-110 It sounds contradictory to state that the crystal structure of LysG had to be solved because there were no suitable homologs, and then to use molecular replacement to solve the structure. Perhaps the authors could tell us retrospectively whether it really was necessary to solve the crystal structure.

Thank you for this excellent question. A homologue suitable for molecular replacement might only share 20% sequence identity if the fold is conserved, but for full atomistic modeling of side chain positions, substantially higher sequence identity would be necessary. This is especially true for predictions regarding effector/ligand binding sites. For LysG, no homologue could be found that would have yielded a sufficiently precise model (the closest related structure (ArgP) shares around 40% identical residues). This study aims at the discussion of detailed ligand interactions, for which a crystal structure is a necessary starting model in absence of an excellent homology model. In addition, the structure with bound effector strengthens the discussion by delivering experimental evidence.

In response to this particular question of the reviewer, we calculated a LysG homology model based on the ArgP-structure. While the overall fold is well represented of course, the modelled binding site geometry suffers from backbone shifts of up to 1-2Å and varying sidechain orientations. This would not have been a good starting point for the selection of residues to be targeted by multi-site directed mutagenesis or any simulation.

→ **To avoid any potential confusion, we rephrased the sentence:** *“As no structure of LysG of *C. glutamicum* was available, we set out to solve the structure of full length LysG with and without basic amino acids as ligands by single-crystal X-ray crystallography to obtain a sound experimental basis for analysis of effector binding.”* (page 5, line 111ff).

8. In this same paragraph, one can't help wondering why the structure of LysG wasn't predicted using David Baker's approach of coupling the sequences of a large number of homologs to a prediction by Rosetta.

Even though computational protein structure prediction and homology modeling have recently become more reliable, at the time when this experiment was performed, it was not conceivable to obtain a structure of LysG with sufficient detail from homology modeling. In absence of sufficiently high sequence identity of solved structures and with the aim to obtain a high-resolution binding site conformation, the experimental structure determination was absolutely necessary (see also the answer to comment 4 of Reviewer #3).

→ **We included the notion on experimental structural information in the manuscript as discussed above (see B).**

10. And now, the arguments for removal of all mention of docking and molecular dynamics. With respect to docking. First note the authors' statement at the end of the legend of Fig. S 3. "We find that the docking tools do not agree very well amongst each other, and none of the findings provide a conclusive argument for alternative binding poses for L-histidine." Such a statement invalidates the statement made elsewhere in the paper relating to the number of water molecules released from the protein upon one ligand's binding. The precise use of the docking programs is not described and it does not appear that others in the field could exactly reproduce what the authors claim. Also, if such docking programs are effective, I would expect them to be used to find and improve potential drugs against medically important proteins. Are they?

This is an interesting remark. The current literature has many prominent examples of successful application of docking programs in in-silico drug discovery procedures. Indeed, within Pharma industry, a strong shift towards docking based library screening can be observed. Below are some insightful review papers from the last 4 years that provide ample evidence for the scientific validity of such tools. It is also important to highlight that our study combines docking with MD simulations and that the ligands have not diffused substantially from the proposed binding site during the relatively long simulation times. This indicates that Docking Scoring Functions, as well as Classical Protein Force Fields align in this study.

Ahsan, M. J., Choupra, A., Sharma, R. K., Jadav, S. S., Padmaja, P., Hassan, M., ... & Bakht, M. A. (2018). Rationale design, synthesis, cytotoxicity evaluation, and molecular docking studies of 1, 3, 4-oxadiazole analogues. *Anti-Cancer Agents in Medicinal Chemistry (Formerly Current Medicinal Chemistry-Anti-Cancer Agents)*, 18(1), 121-138.

Pagadala, N. S., Syed, K., & Tuszynski, J. (2017). Software for molecular docking: a review. *Biophysical reviews*, 9(2), 91-102.

De Vivo, M.; Masetti, M.; Bottegoni, G.; Cavalli, A. Role of Molecular Dynamics and Related Methods in Drug Discovery. *J. Med. Chem.* 2016, 59, 4035–4061.

→ In response to this remark we now removed Supplementary Fig S13 and the corresponding legend. We also made the scripts for rerunning the docking calculations available in the persistent repository SimTK (see A.3 above).

11. Finally, the paper would not be weakened by the omission of all mention of docking. Why include the inconclusive docking material when the authors have valid and conclusive experimental data?

Thank you for this suggestion. We discussed this but came to the conclusion that our current claims are in alignment with the level of credibility that can be attributed to docking programs. In order to start an MD simulation, it is necessary to define a starting configuration of the protein+ligand system. In absence of crystal structures for LysG+Lys and LysG+His, it is necessary to place the ligands in a reasonable starting position. Even though docking tools might not place a ligand exactly in the correct position, the rmsd variance between the best scoring docking poses of the used docking tools was very small. In addition, the MD simulations did not diffuse the ligands away from the proposed docking site, providing additional evidence for the correct identification of the binding site.

Given that Docking was solely used to identify the position of the substrates in the binding site, and that it is a necessary part of our MD method, we did not remove this content.

→ In response to this statement we removed the consensus docking method and only focus on a simpler docking approach with one docking tool. (This resulted in the removal of Supplementary Fig SI3 and accompanying legend, see B).

11. (it should be 12, was also 11 in the original revision) *With respect to molecular dynamics. I believe that most folks think that relevant conformational changes in allosteric proteins take place on a time scale of milliseconds.*

Thank you for this remark. Given that the simulations start with the ligand in the active site of the protein, we do not need to simulate the diffusion of the ligand into the active site. Instead, this study focusses on the conformational changes induced by the binding event, which occur on the nanosecond timescale. The best evidence for this is that these transitions were frequently observed during the various simulations as reported here. This is quite common for small conformational changes. Larger conformational changes, like ion channel openings or folding of globular proteins occur on the millisecond timescale instead. Our findings of relevant protein motion on the nanosecond timescale are in agreement with our expectation and current consent in the field. This article (in particular figure 1 therein):

Dror, Ron O., Jensen, Morten Ø., Borhani, David W., Shaw David E. (2010). Exploring atomic resolution physiology on a femtosecond to millisecond timescale using molecular dynamics simulations. *J Gen Physiol.* 2010 Jun; 135(6): 555–562
(<https://www.ncbi.nlm.nih.gov/pmc/articles/PMC2888062/>)

→ In response to this sentence, we added the following sentence: “While allosteric transitions frequently occur on longer timescales, the observed transitions are impacted by ligand binding at the hinge region of the two-lobe regulator domain. It should be noted that the simulation does not describe the allosteric conformational change. The simulation only considers the conformational motion of the two-lobe regulator domain and how it is influenced by different ligands. The resulting allosteric effect is the activation of the transcription factor in the DNA-binding domain which happens further away as a consequence of this domain motion and is not studied here.” (page 17, line 389ff). This is also the response to comment C.2 (see above).

16. *As in the case of docking, I hope the authors will remove this component of their paper in favor of their solid and convincing experimental data.*

We appreciate the thoughtful comments and suggestions throughout. We believe that the additional clarifications and explanations will make this manuscript more accessible to a broader audience. Given that the atomistic behavior observed in the MD provides the best accessible model to explain the experimental data, we kindly disagree with this particular suggestion and prefer to retain the MD data in the text.

→ With this second revision, we did provide additional discussion to contrast the observed behavior with allosteric timescales reported elsewhere. With this additional information, we consider the MD part to be a valuable addition to the reported findings of this study.

Reviewer #4

I satisfy the answers from the authors.

-Thank you!

REVIEWERS' COMMENTS:

Reviewer #3 (Remarks to the Author):

The paper has been improved by the removal of the use of the multiple docking programs and the inclusion of more details on the molecular dynamics calculations. I remain unconvinced however of the need for the molecular dynamics in this paper and unconvinced that the molecular dynamics mean anything. I suggest that the authors remove the MD from this paper and submit this part of the work separately to a specialist journal where multiple experts on the topic can offer their opinions.

Point-by-point response to Reviewers

Manuscript ID: NCOMMS-20-10185B

Engineering and application of a biosensor with focused ligand specificity

Reviewer #3 (Remarks to the Author):

1. The paper has been improved by the removal of the use of the multiple docking programs and the inclusion of more details on the molecular dynamics calculations. I remain unconvinced however of the need for the molecular dynamics in this paper and unconvinced that the molecular dynamics mean anything. I suggest that the authors remove the MD from this paper and submit this part of the work separately to a specialist journal where multiple experts on the topic can offer their opinions.

We deeply regret that we could not convince Reviewer #3 with the revision of our manuscript. Nevertheless, we would like to thank Reviewer #3 for the helpful suggestions during the reviewing process as they helped us to increase the readability of the manuscript.